# Senescence-associated lineage-aberrant plasticity evokes T-cell-mediated tumor control

Dimitri Belenki [1,2], Paulina Richter-Pechanska[1], Zhiting Shao[1], Animesh Bhattacharya[1], Andrea Lau[1], José Américo Nabuco Leva Ferreira de Freitas [3], Gregor Kandler[1], Timon P. Hick[1,4], Xiurong Cai[1], Eva Scharnagl[5], Aitomi Bittner [1], Martin Schönlein[1,6,7], Julia Kase[1], Katharina Pardon[1], Bernadette Brzezicha[8], Nina Thiessen[9], Oliver Bischof [3], Jan R. Dörr[1,10,11], Maurice Reimann[1], Maja Milanovic [1,12,13], Jing Du[14], Yong Yu [5], Björn Chapuy [12], Soyoung Lee [1,5], Ulf Leser[4], Claus Scheidereit [2], Jana Wolf [2,15], Dorothy N. Y. Fan[1,2,13] & Clemens A. Schmitt [1,2,5,13,16] ✉

Cellular senescence is a stress-inducible state switch relevant in aging, tumorigenesis and cancer therapy. Beyond a lasting arrest, senescent cells are characterized by profound chromatin remodeling and transcriptional reprogramming. We show here myeloid-skewed aberrant lineage plasticity and its immunological ramifications in therapy-induced senescence (TIS) of primary human and murine B-cell lymphoma. We find myeloid transcription factor (TF) networks, specifically AP-1-, C/EBPβ- and PU.1-governed transcriptional programs, enriched in TIS but not in equally chemotherapy-exposed senescence-incapable cancer cells. Dependent on these master TF, TIS lymphoma cells adopt a lineage-promiscuous state with properties of monocytic-dendritic cell (DC) differentiation. TIS lymphoma cells are preferentially lysed by T-cells in vitro, and mice harboring DC-skewed Eμ-*myc* lymphoma experience significantly longer tumor-free survival. Consistently, superior long-term outcome is also achieved in diffuse large B-cell lymphoma patients with high expression of a TIS-related DC signature. In essence, these data demonstrate a therapeutically exploitable, prognostically favorable immunogenic role of senescence-dependent aberrant myeloid plasticity in B-cell lymphoma.

Cellular senescence reflects a unique state switch of cellular functionalities beyond an apoptosis-comparable terminal cell-cycle exit[1,2]. Most prominently, senescent cells present with a largely pro-inflammatory and matrix-active secretome, termed the senescence-associated secretory phenotype (SASP)[3], and attract immune cells that, at least in part, clear senescent cells[4–6]. Physiological roles of senescence during embryogenesis and wound healing cannot simply be explained by a lasting cell-cycle cessation[7–10], but imply senescence-associated plasticity to promote organ development or to restore tissue integrity by replenishing parenchymal cellularity[1,11,12]. We and others previously described senescence-associated stem cell-like reprogramming as a result of instructive SASP effects on neighboring non-senescent cells, or even as an intrinsic event in epigenomically remodeled senescent cells that subsequently managed to resume proliferation as post-senescent cells[10,13–24]. More specifically, we have recently linked enhancer chromatin remodeling and transcription

factor (TF) recruitment to senescence dynamics, and identified the activator protein 1 (AP-1) as a pioneering top-hierarchical TF that orchestrates transcriptional programs in senescence, and further controls lasting chromatin changes in senescence-escaped cells[25,26].

Cancer cells are known to exploit developmental programs and lineage promiscuity, thereby gaining growth competitiveness and drug resistance[27–30]. For instance, lineage plasticity reportedly underlies neuroendocrine differentiation in lung and prostate cancer[31], and cross-differentiation (a term used here to distinguish from complete transdifferentiation of one cell type into another) towards myeloid and T-cell markers represents a hallmark feature of classic Hodgkin's lymphoma[32,33]. Reflecting actual transdifferentiation, B-lymphocytes can be reprogrammed into macrophages (Mφ) via engineering of the TF CCAAT-enhancer-binding protein (C/EBP) and subsequent involvement of PU.1 (a.k.a. SPI1)[34]. Despite the intimate relationship between cellular senescence and plasticity, little is known about the functional implications of cross-differentiation related to senescence induction[15], especially by anti-cancer treatment[35]. We explore here aggressive B-cell lymphoma biology regarding senescence-associated plasticity in the Eµ-*myc* transgenic mouse B-cell lymphoma model. This model was reportedly instrumental in unveiling fundamental components of senescence susceptibility in cross-species analyses of diffuse large B-cell lymphoma (DLBCL) patient data sets, further supported by investigations of DLBCL cell lines and primary patient material[6,16,25,36–39]. We identify here key TF, namely AP-1, C/EBPβ, and PU.1[40–42], all previously linked to lineage-aberrant phenotypes in hematologic malignancies[32,43–45], as main drivers of senescence-associated myeloid-biased cancer cell cross-differentiation. Our findings presented here functionally connect T-cell-mediated immunosurveillance of senescence-associated dendritic cell (DC)-like plasticity to superior long-term outcome.

## Results

### B-cell lymphomas exhibit monocytic-DC identity markers in TIS

We utilized primary Eµ-*myc* transgenic mouse B-cell lymphomas with and without genomic ablation of the senescence-essential H3K9 methyltransferase *Suv39h1* and further engineered to overexpress Bcl2 as a well-established model for therapy-induced senescence (TIS)[16,25,36–39,46] (Supplementary Fig. 1a). Since lineage plasticity relies on differential TF activities and their corresponding transcriptional output, we first explored which TF regulate the TIS-associated lymphoma transcriptome that we previously reported after in vitro-exposure to the anthracycline chemo drug adriamycin (ADR, a.k.a. doxorubicin)[39], encompassing transcripts that distinguish senescent Eµ-*myc;bcl2* (hereafter referred to as control;*bcl2*) from equally ADR-treated but senescence-incapable *Suv39h1*-deficient Eµ-*myc;bcl2* (i.e. *Suv39h1⁻;bcl2*) lymphomas and their respective untreated matches. In addition to NF-κB-controlled SASP factors, a gene-regulatory interaction map generated by induced network module analysis[47] uncovered a large proportion of TIS-specific transcripts as regulatory targets of the TF AP-1, C/EBPβ, or PU.1 (Fig. 1a, Supplementary Data 1). Gene set enrichment analyses (GSEA[48]) revealed a significant enrichment of previously reported target gene sets of these three TF[25,49,50], suggesting their senescence-induced hyperactivity in TIS (Supplementary Fig. 1b). Capillary immunoassay of nuclear lymphoma lysates regarding expression of AP-1 (with a focus on its components JunD and c-Jun, the latter also in its S63-phosphorylated, derepressed form[51]), C/EBPβ and PU.1 detected higher fold-induction of all three TF proteins in TIS compared to senescence-impaired lymphomas (Fig. 1b). Increased activity of these TF, as inferred by GSEA-exploited target gene expression, was also evident in other TIS models of hematopoietic and solid malignancies, in Braf^V600E- and Hras^G12V-senescent melanocytes and human diploid fibroblasts (HDF) as examples of oncogene-induced senescence (OIS)[52], in Pten loss-induced cellular senescence (PICS) of hyperplastic prostate epithelium[53], as well as in virus-induced senescence[54] (Supplementary Fig. 1c). Thus, these findings mark elevated activities of the three TF across various types of cellular senescence, especially in senescence-susceptible tumors upon cancer therapy.

Since PU.1 establishes a Mφ-specific enhancer landscape through collaboration with C/EBP and AP-1 factors[55], we next asked whether the TIS state switch was associated with lineage infidelity in our B-cell lymphomas. First, interrogation of the 'Haemosphere' database found the top-100 TIS-associated genes predominantly linked to myeloid cell lineages[56] (Fig. 1c). Probing TIS control;*bcl2* lymphomas compared to equally ADR-exposed *Suv39h1⁻;bcl2* samples by GSEA unveiled strong skewing towards various myeloid gene sets, including DC and Mφ profiles, a facet shared with other TIS, OIS, PICS and VIS models (Supplementary Fig. 1b–d). Next, we obtained single-cell RNA sequencing (scRNA-seq) profiles from lymphoma cells in which 4-OH-tamoxifen (4-OHT) conditionally restored senescence capacity by activating a Suv39h1-estrogen receptor fusion in a Suv39h1-deficient background[16], and found a strongly TIS-associated expression of DC, PU.1 and other myeloid-related transcriptomic signatures on the scRNA level (Fig. 1d; see related bulk RNA profiles in Suv39h1-proficient vs. -deficient settings in Supplementary Fig. 1d). Intriguingly, a sizeable fraction of the senescent lymphoma cells not only presented with a DC-like but simultaneously an adult tissue stem cell (ATSC) signature[57], which we previously linked to senescence-associated stemness[16], demonstrating that senescence-associated plasticity can aberrantly co-exhibit stem-like and cross-differentiation features in the same cells (Fig. 1d). Remarkably, we noted a shift of the neoplastic B-cells towards monocytoid morphology (with the ANA-1 Mφ cell line as a reference) exclusively in ADR-exposed senescence-capable lymphomas (Fig. 1e), and detected intermediate expression levels of myeloid TF compared to ANA-1 cells and non-senescent lymphomas (Supplementary Fig. 2a). We then analyzed expression of monocytic surface markers by flow cytometry and detected a TIS-associated gain of CD11b (integrin alpha M [ITGAM]) or CD115 ([macrophage-]colony stimulating factor [CSF] receptor[-1]), abbreviated as M-CSFR or CSF1R) surface markers on lymphoma subpopulations in response to ADR, γ-irradiation (IRR) or the CDK4/6 inhibitor palbociclib (palbo; Fig. 1f, Supplementary Fig. 2b, c). While the expression of other myeloid-lineage related molecules such as CD11c (ITGAX) or Fc gamma III/II receptors (CD16/32) was also enhanced on TIS cells, there was no concomitant loss but rather a gain of B-cell marker expression, such as CD19 and the PU.1-controlled B220 marker (Supplementary Fig. 2d, e). Notably, a series of GSEA on activated vs. resting B-cells and TIS vs. non-TIS cells concluded that the senescence-associated myeloid shift cannot be explained by B-cell activation (Supplementary Fig. 2f, g). Hence, different from advanced or fully completed transdifferentiation[34], senescent cells display aberrant plasticity with a lineage-promiscuous marker profile.

Because the catalytic subunit of the NADPH oxidase complex, CYBB (also referred to as NOX2 or gp91^phox), is a PU.1 target (Fig. 1a), we quantified the respiratory burst capacity in Eµ-*myc;bcl2* lymphoma cells using a p-nitroblue tetrazolium (NBT) reduction assay[58]. Indeed, lymphomas that entered TIS upon ADR or palbo exposure, but not equally treated Suv39h1-deficient lymphomas, displayed profound respiratory burst activity, reminiscent of monocytoid cells able to engulf neighboring cells[59] (Fig. 1g; compared to ANA-1 cells). The TIS-associated gain of respiratory burst activity was further recapitulated in Hras^G12V-senescent HDF and -senescent mouse embryonic fibroblasts (MEF) (Supplementary Fig. 3a-d). These findings demonstrate that B-cell lymphoma senescence comes with a cellular state switch that includes lineage-inappropriate features reminiscent of a monocytic, specifically a DC- or Mφ-like cross-differentiation phenotype.

### Monocyte-DC identity in TIS is linked to a myeloid-skewed TF repertoire

Next, we investigated whether aberrant TF activities accompany the B-lymphoid-to-myeloid (B-to-M) transgression. Because CD115 marks

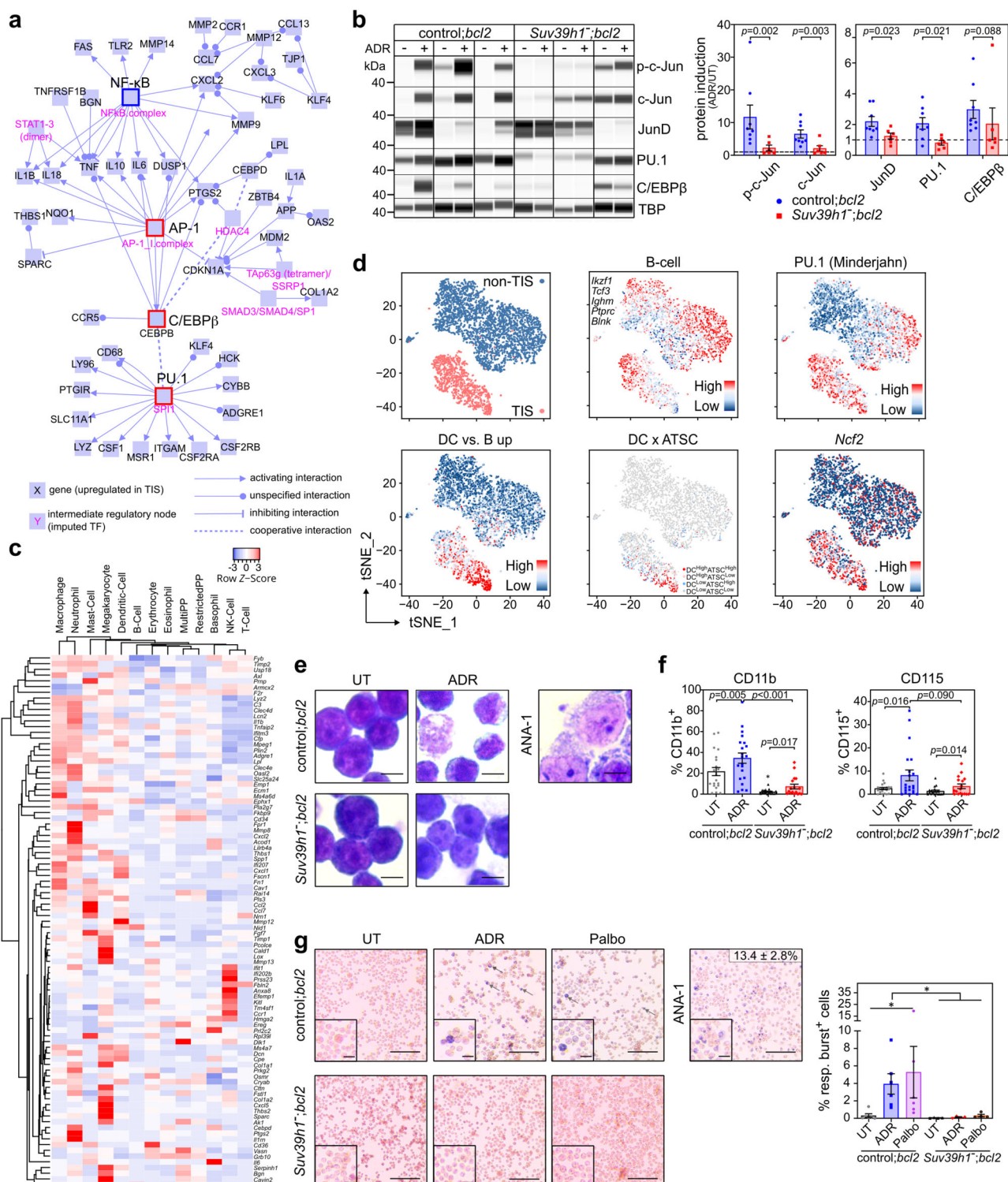

more specialized monocytic cells than the pan-myeloid marker CD11b, we focused on the CD115-positive subpopulation of TIS control;*bcl2* lymphomas, which we hypothesized to reflect a more advanced stage in the B-to-M transition compared to the CD115⁻ fraction (Supplementary Fig. 4a). Consistent with the senescent state switch underlying the acquisition of myeloid features, a significantly higher proportion of sorted CD115⁺ as compared to CD115⁻ B-cell lymphoma cells presented as senescence-associated β-galactosidase (SA-β-gal) marker-positive[60] (Fig. 2a). Importantly, CD115⁺ cell fractions displayed more abundant C/EBPβ and PU.1 TF protein expression, while expression of the B-cell master regulator PAX5 remained largely unchanged (Fig. 2b).

Quantitative real-time reverse transcriptase PCR (qRT-PCR) corroborated the CD115-related B-to-M conversion on the transcriptional level, as highlighted by increased expression of myeloid lineage-related TF and target genes, slightly decreased abundance of B-cell identity-mirroring transcripts, and more pronounced stem cell markers in the CD115⁺ TIS fraction (Fig. 2c). Of note, we interrogated immune deconvolution algorithms, namely xCell, quanTIseq or CIBERSORT[61–63], that were developed to decipher contributing cell types from RNA-seq and microarray data sets. Remarkably, all these tools assigned high myeloid scores, indicative of DC or Mφ cell fractions, to TIS vs. untreated transcriptomes obtained from pure B-cell

**Fig. 1 | Senescent B-lymphoma cells exhibit phenotypic and functional myeloid plasticity. a** Induced gene-regulatory network[47] of upregulated genes from the Eμ-*myc;bcl2* therapy-induced senescence (TIS) model[39]. Red boxes represent interconnected regulatory hubs consisting of AP-1, PU.1 and C/EBPβ transcription factors (TF); blue box marks a TIS-related NF-κB hub. **b** Capillary-based immunoassay of hematopoietic TF in nuclear lysates of control;*bcl2* (n = 8) and *Suv39h1⁻;bcl2* (n = 6) lymphomas, 5-day exposed to adriamycin (ADR) or left untreated (UT) (left). TATA box-binding protein (TBP) as loading control. Representative digital blots of three lymphomas per genotype. Mean TF fold-induction values ± SEM compared to matched UT lymphomas (right). **c** Hierarchical clustering of top 100 TIS-upregulated genes as in **a** (93 measured and shown), for their expression in hematopoietic lineages from 'Haemopedia'[56]. **d** tSNE clustering of scRNA-seq profiles from conditionally senescent (TIS, red dots) and non-senescent (non-TIS, blue dots) Suv39h1-inducible lymphomas[16] (top left); specific analyses of B-cell transcripts, enrichment scores of PU.1 and dendritic cell (DC), adult tissue stem cell signature (ATSC) previously associated with senescence[16,57], and *Ncf2* expression as indicated. **e** Representative May-Gruenwald-Giemsa stain (n = 5 lymphomas per genotype); ANA-1 macrophages for comparison. Scale bars, 5 μm. **f** Flow cytometry-determined percentage of CD11b⁺ (left) and CD115⁺ (right) control;*bcl2* (n = 21) and *Suv39h1⁻;bcl2* (n = 17) lymphomas as in **b**. **g** Respiratory burst of control;*bcl2* (n = 6) and *Suv39h1⁻;bcl2* (n = 4) lymphomas 5 days after ADR or palbociclib (Palbo) treatment or left UT. Representative microscopy images (left) and quantification (right). Arrows indicate reduced nitroblue-tetrazolium (NBT) precipitates. Scale bars, 100 μm or 20 μm for inlays. ANA-1-macrophages (middle) with the percentage of reduced NBT⁺ cells ± SD (n = 3 technical replicates). **f, g** Bars indicate the mean percentage of positive cells ± SEM. Two-sided *P*-values by unpaired *t*-test on log₂-transformed data (**b**); by paired *t*-test for UT vs. ADR comparisons of the same genotype and unpaired *t*-test for control vs. *Suv39h1⁻* comparisons (**f**). * two-sided *P* by non-parametric Friedman test comparing senescent to UT conditions (P = 0.006) or by two-way ANOVA assessing the contribution of *Suv39h1⁻* genotype to the data variance (P = 0.032) (**g**). Source data are provided in the Source Data file.

---

lymphoma isolates, despite their common B-cell, not myeloid, origin (Fig. 2d, Supplementary Data 2), thus further underscoring the aberrant myeloid cross-differentiation phenotype of senescent B-cell lymphoma cells.

## Aberrant differentiation in TIS is co-regulated by PU.1, C/EBPβ and AP-1

To mechanistically demonstrate that lineage-aberrant TF activity drives lineage plasticity of TIS lymphomas, we depleted PU.1/SPI1, C/EBPβ and AP-1 (c-Jun) by shRNA-mediated gene knockdown (Supplementary Fig. 4b). Despite largely unchanged senescence susceptibilities (Supplementary Fig. 4c), TIS-associated CD115 and CD11b surface expression levels as well as respiratory burst capacities were suppressed in settings with reduced *Spi1* and *Cebpb* expression (Fig. 3a, b), while *Jun* depletion produced no such effect (Supplementary Fig. 4d). To investigate the global impact of the three TF on gene expression, we performed RNA-seq analysis of untreated vs. TIS lymphomas with respective TF depletions. Since PU.1, C/EBPβ and AP-1 reportedly cooperate to drive expression of target genes such as *Il1b* and *Csf1r* [64,65], we also probed potential cross-regulatory effects on differential expression (DE) by depletion of any individual TF on target gene sets of the other two TF, using GSEA (Supplementary Fig. 4e, Supplementary Data 3). GSEA results were in agreement with the mode of cooperative TF action (Supplementary Fig. 4f). Mutual regulation of target genes was also evident when we analyzed the chromatin of TIS lymphomas in the Suv39h1 restoration model by Assay for Transposase-Accessible Chromatin using sequencing (ATAC-seq) (Supplementary Fig. 4g). Although there was no considerable impact on myeloid marker expression upon *Jun* knockdown, we found the dominant-negative c-Jun moiety cJunΔN, which is able to disrupt activity of all JUN containing AP-1 dimers[66], to reduce expression of the three TF, and also of myeloid lineage transcripts in TIS lymphomas (Supplementary Fig. 4b, h). Therefore, we generated lymphomas with genetic impairment of all three TF, leading to most profound suppression of myeloid transcripts in the TIS state when compared to sole disruption of any single factor (Fig. 3c). Collectively, these data pinpoint cooperative regulation of cellular plasticity by a perturbed myeloid differentiation-governing core TF network.

As revealed by GSEA, the induction of the Mφ gene signature was reduced in TIS lymphomas upon PU.1 or C/EBPβ depletion, with *Spi1* knockdown also neutralizing the DC-like signature in TIS (Fig. 3d, Supplementary Fig. 4i). Moreover, the aforementioned TIS-related myeloid cell type assignments by immune deconvolution algorithms were largely lost by *Spi1* knockdown (Supplementary Fig. 4j, Supplementary Data 2). Conversely, we detected elevated levels of myeloid lineage-related PU.1 target genes such as *Il1b, Csf1r*, or *Csf2rb2* when inducing senescence under concurrent activation of PU.1-ER^T2, a modified estrogen receptor (ER^T2) fusion protein selectively responsive to 4-OHT (Fig. 3e). Hence, PU.1 is particularly important to instruct the transition towards a Mφ-DC state in senescent neoplastic B-cells.

## Human DLBCL cells acquire aberrant myeloid differentiation features in TIS

To validate these findings in human B-cell lymphomas, we first classified a panel of 18 DLBCL cell lines by their ADR-inducible SA-β-gal-based senescence responsiveness as either senescence-competent, -intermediate or -incompetent (Fig. 4a, Supplementary Fig. 5a). Only ADR-exposed senescence-competent cell lines presented with a substantial oxidative burst capacity, comparable to human THP-1 monocytoid leukemic cells (Fig. 4b, Supplementary Fig. 5a), and induced the DC marker CD11c (Fig. 4c, Supplementary Fig. 5a).

Next, we performed TIS-focused RNA-seq of the 18 DLBCL cell lines. ADR-treated cells were prelabeled with a proliferation-tracking dye (CellTrace FarRed) and enriched for senescent, cell-cycle-arrested cells by flow-sorting of CellTrace^high cells, while treatment-naïve samples were depleted of senescent or slowly proliferating cells by sorting for CellTrace^low cells, thereby maximizing the contrast of senescence-attributable gene expression profiles (GEP; Supplementary Fig. 5b, Supplementary Data 4). We conducted a TF footprint analysis on the DLBCL RNA-seq data using the DoRothEA tool[67]. Among the top-50 differentially regulated TF activities induced by ADR treatment we found – in addition to NF-κB and STAT factors – JUN, CEBPB, and SPI1 when comparing the senescence-competent vs. -incompetent and JUN and SPI1 when comparing senescence-competent vs. senescence-intermediate cell lines (Fig. 4d, Supplementary Fig. 5c). *STAT3, ITGAX* (encoding CD11c) and the neutrophil oxidative burst mediator *NCF2* were marked as particularly pronounced interaction partners in SPI1 regulons selectively in ADR-exposed senescence-competent human cell lines (Supplementary Fig. 5d; see also Fig. 1d). Collectively, these findings suggest the presence of a TIS-associated, permissive chromatin environment at loci of the target genes of the respective TF. Accordingly, myeloid-indicative *ITGAX* and *CSF1R* transcripts as well as detectable surface CD11b were most prominently upregulated in senescence-competent TIS cells (Supplementary Fig. 5e, f). Finally, immune deconvolution algorithms detected the pseudo-presence of myeloid cell types most profoundly in senescence-competent ADR-exposed DLBCL lines (Supplementary Fig. 5g, Supplementary Data 2). Thus, a strong TIS response of human DLBCL is also characterized by a marked B-to-M transition as reported above for senescent murine lymphomas.

## PU.1, C/EBPβ and JUN regulate TIS and lineage fidelity in DLBCL

In line with the functional role of the three master TF in B-cell lymphoma TIS, enforced expression of ER^T2-fused TF in senescence-competent DLBCL cell lines Karpas-422 and RC-K8 (Supplementary Fig. 6a) slowed proliferation, enhanced the senescence response (Fig. 4e,

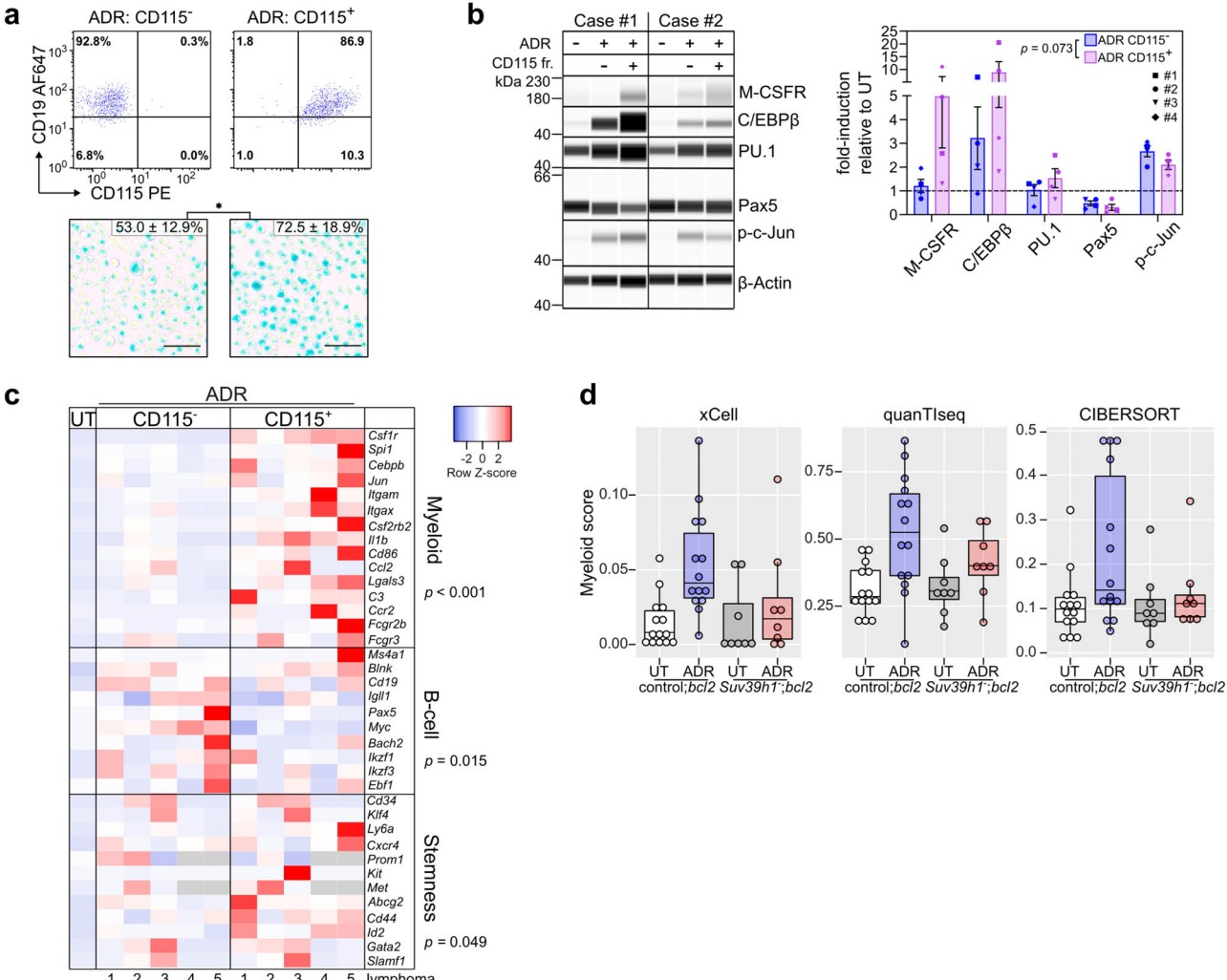

**Fig. 2 | Lineage-infidel TIS cell populations exhibit a myeloid-skewed transcriptional profile. a** Individual control;*bcl2* lymphomas (*n* = 5) were exposed to adriamycin (ADR) for 5 days and flow-sorted into CD115⁺ vs. CD115⁻ populations (top, with CD19 as a co-staining; quadrant gate percentages shown), then stained for senescence-associated β-galactosidase (SA-β-gal). Representative microscopy images and quantification results as mean percentage of positive cells ± SEM (bottom). Scale bars, 50 µm. * *P* = 0.025 by two-sided, paired *t*-test. **b** Capillary-based immunoassay of hematopoietic TF in whole-cell lysates of individual control;*bcl2* lymphomas (*n* = 4), left untreated (UT) or flow-sorted into cell fractions (fr.) based on CD115 surface expression as in **a**, but 7 days after ADR exposure and two days after sorting. Digital blots for two representative lymphomas (left, β-Actin as loading control) and quantification of protein expression as mean fold-change values ± SEM relative to matched UT samples and normalized to β-Actin (right).

Each dot in the plot represents an individual lymphoma as indicated. **c** Heatmap representation of transcript abundance measured by qRT-PCR in samples as in **b** (*n* = 5), with fold-change values relative to matched UT samples as row *Z*-scores. Transcripts are grouped into myeloid, B-cell, and stemness[16] groups (cf. Source Data). **b**, **c** *P*-values from fixed-effects linear models reflect the statistical significance of the CD115 factor, see Methods for details. **d** Myeloid scores generated by the indicated immune deconvolution algorithms on microarray-based transcriptome profiles of ADR-treated or drug-naïve (UT) Eµ-*myc*;*bcl2* lymphomas (*n* = 14 for control and *n* = 8 for *Suv39h1*⁻ genotypes (cf. Supplementary Data 2). Box plots show the median as center line, 25% and 75% percentiles as box limits and whiskers extending to the largest and smallest values within the 1.5× interquartile range of the box limits. Source data are provided in the Source Data file.

Supplementary Fig. 6b), and induced CD11c expression most strongly when 4-OHT was co-administered with ADR (Fig. 4f, Supplementary Fig. 6c). Except for a decreased proliferation rate, no senescence or CD11c induction was evident in the equally engineered and treated senescence-incapable SU-DHL5 cell line (Supplementary Fig. 6d).

Further corroborating these results, GEP by qRT-PCR of TF-induced and treatment-exposed Karpas-422 cells marked a strong induction of SASP transcripts and NF-κB targets as well as myeloid lineage transcripts with variable but consistent effects (regarding their target gene profiles) across single TF (Fig. 4g). Taken together, these results pinpoint PU.1, C/EBPβ and JUN as drivers of partly overlapping and partly distinct senescence-sensitive lineage plasticity programs in DLBCL cells, unveiling a causal relationship between B-cell-inappropriate TF levels, senescence and myeloid lineage features.

## The TIS-associated myeloid switch impacts on long-term lymphoma outcome

To explore the prognostic power of TF-governed lineage-aberrant plasticity, we first interrogated the treatment outcome of a mouse cohort bearing primary, individual Eµ-*myc* (and otherwise non-engineered) lymphomas. In this clinical trial-like setting, lymphomas were classified at diagnosis, based on their actual subsequent responses to a single-dose of cyclophosphamide (CTX) chemotherapy in vivo, as either curable (i.e. never relapsing [NR]) or relapsing (i.e. relapse-prone [RP])[39]. Transcriptome data were available from bulk lymph-node lysates at baseline (i.e. prior to therapy) and after a 4-hr short-term CTX exposure in vivo. GSEA unveiled a strong skewing towards myeloid and TF gene sets in GEP of short-term CTX-challenged NR vs. RP lymphomas (Fig. 5a), thereby connecting these gene modules to superior

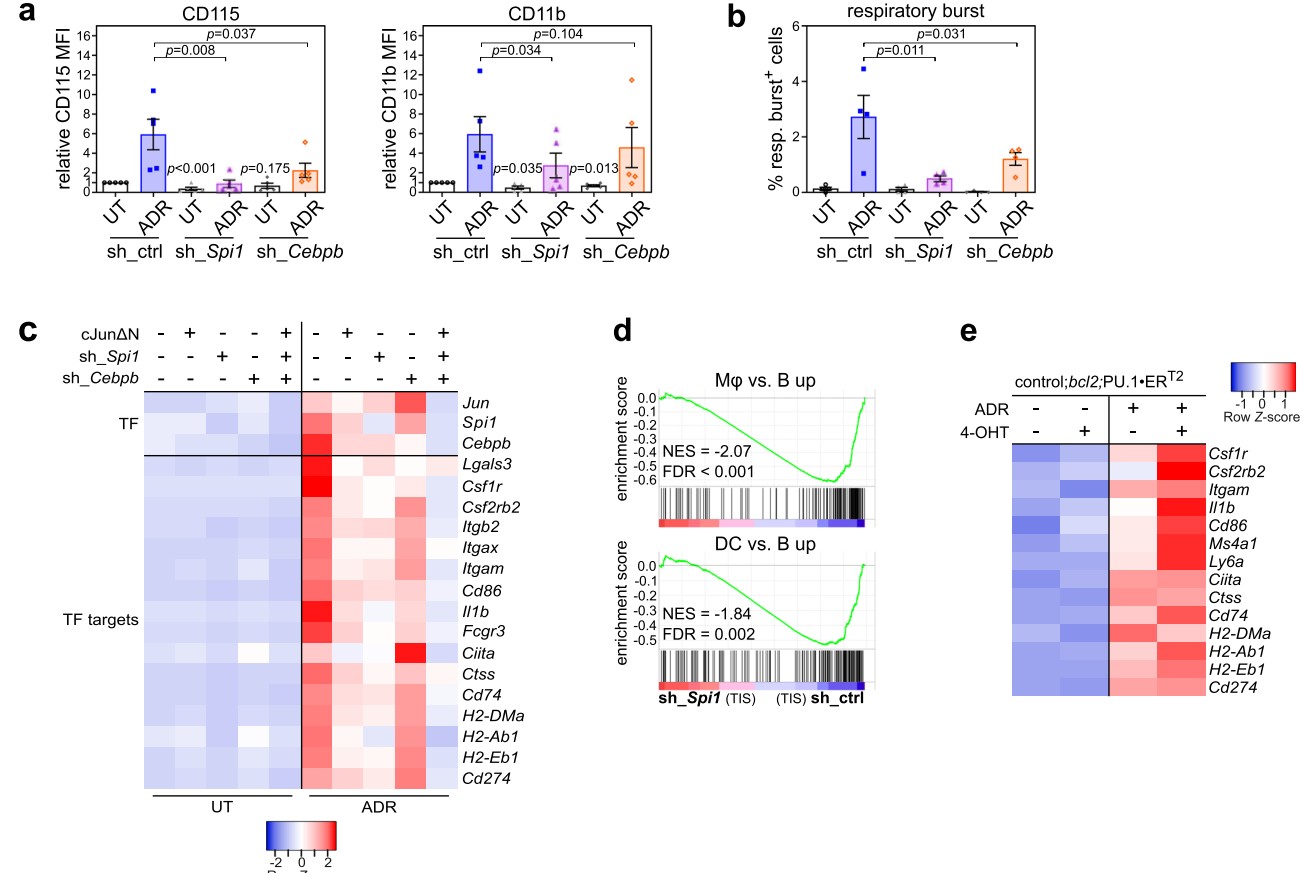

**Fig. 3 | PU.1, C/EBPβ, and AP-1 TF cooperatively regulate aberrant lineage transition. a** Flow-cytometric analyses of CD115 and CD11b expression in biologically individual untreated (UT) vs. 5-day adriamycin (ADR)-exposed control;*bcl2* lymphomas (*n* = 5 each), stably transduced with either a control (sh_ctrl) shRNA or shRNAs targeting *Spi1* (sh_*Spi1*) or *Cebpb* (sh_*Cebpb*). Mean fluorescence intensities (MFI) relative to UT sh_ctrl are plotted. **b** Quantification of the respiratory burst capacity by NBT assay of control;*bcl2* lymphomas, as in **a** (*n* = 4 each). **a, b** Bars indicate the mean fold-expression (**a**) or mean percentage of positive cells (**b**) ± SEM. *P*-values by two-sided ratio paired (comparisons indicated by connecting lines) or by two-sided one-sample *t*-test (comparing UT sh_TF to UT sh_ctrl). **c** Heatmap of qRT-PCR-based gene expression profiles comparing UT vs. 5-day ADR-exposed control;*bcl2* lymphomas transduced with either control vector,

cJunΔN, sh_*Spi1*, sh_*Cebpb*, or the combination of all three latter constructs (*n* = 3 individual lymphomas each). Analyzed transcripts encompass the directly disrupted TF and their target genes. **d** RNA-seq-based gene set enrichment analysis (GSEA) of individual therapy-induced senescent (TIS) lymphomas with *Spi1* depletion (sh_*Spi1*) compared to sh_ctrl (*n* = 4 each), probing gene sets upregulated in macrophages (Mφ, top) and dendritic cells (DC, bottom). Negative normalized enrichment scores (NES) signify depletion in sh_*Spi1* TIS. False discovery rate (FDR) as indicated. **e** Heatmap presentation of qRT-PCR-based gene expression data of PU.1 target transcripts in individual control;*bcl2* lymphomas stably transduced with a 4-hydroxytamoxifen (4-OHT)-inducible PU.1-ER^T2 fusion construct, upon 6-day ADR exposure or left UT, and simultaneously exposed to 4-OHT or solvent control (*n* = 4). Source data are provided in the Source Data file.

outcomes. Notably, the Mφ gene set was already found enriched in treatment-naïve NR compared to RP tumors (Fig. 5b), indicating a particular susceptibility of primary B-lymphoma cells in their natural in vivo contexts to respond to endogenous oncogenic and environmental stresses in a similar direction as seen under exogenous chemotherapeutic stress. Importantly, the Mφ and DC bias was reproduced in CD19[+] bead-selected, i.e. B-cell-purified cells from curable Eμ-*myc* lymphomas (Fig. 5c, d), and further detected in comparison to normal reactive germinal center B (GCB)-cells (Supplementary Fig. 7a), thereby confirming that the observed myeloid skewing was truly attributable to the neoplastic B-cell population and not its tumor microenvironment (TME).

Based on a large DLBCL patient cohort with publicly available individual transcriptome and annotated clinical course information[68], GSEA indicated enrichment of a human DC gene signature in bulk treatment-naïve DLBCL material from patients that achieved lasting disease control (i.e. who did not decease over extended periods of time) in response to R-CHOP (rituximab, CTX, ADR, vincristine and prednisone) induction therapy (Fig. 5e), confirming our mouse-model based finding. Similarly, GEP of CD19[+]-selected B-cells from primary

DLBCL patient samples were enriched for the DC profile when compared to a normal human GCB centroblast data set (Supplementary Fig. 7b). Hence, cell-intrinsic expression of lineage-inappropriate gene modules is a propensity of malignant B-lymphoid cells shared in an outcome-discriminative fashion across the mouse and human species.

Next, we stratified lymphoma-bearing mice of our clinical trial-like cohort based on their individual lymphoma GEP after a 4-hr CTX in vivo challenge. Plotted in Kaplan-Meier survival format, mice harboring Eμ-*myc* lymphomas expressing a high-level DC signature (i.e. above median expression) achieved significantly longer tumor-free survival (Fig. 5f). Similarly, above median scores of a human DC signature were associated with superior overall survival in two independent DLBCL patient cohorts (Fig. 5g). The DC^high subgroup of DLBCL presented with a significantly lower number of mutations and deletions in senescence gatekeeper gene loci *TP53* or *CDKN2A*[52,69], indicative of DC-like plasticity manifesting preferentially in those DLBCL cases that retained functional senescence capacity (Supplementary Fig. 7c). Notably, although DC^high DLBCL cases associated with higher expression of genes related to MHC class I antigen presentation, the above median expression of these genes alone was not associated with superior

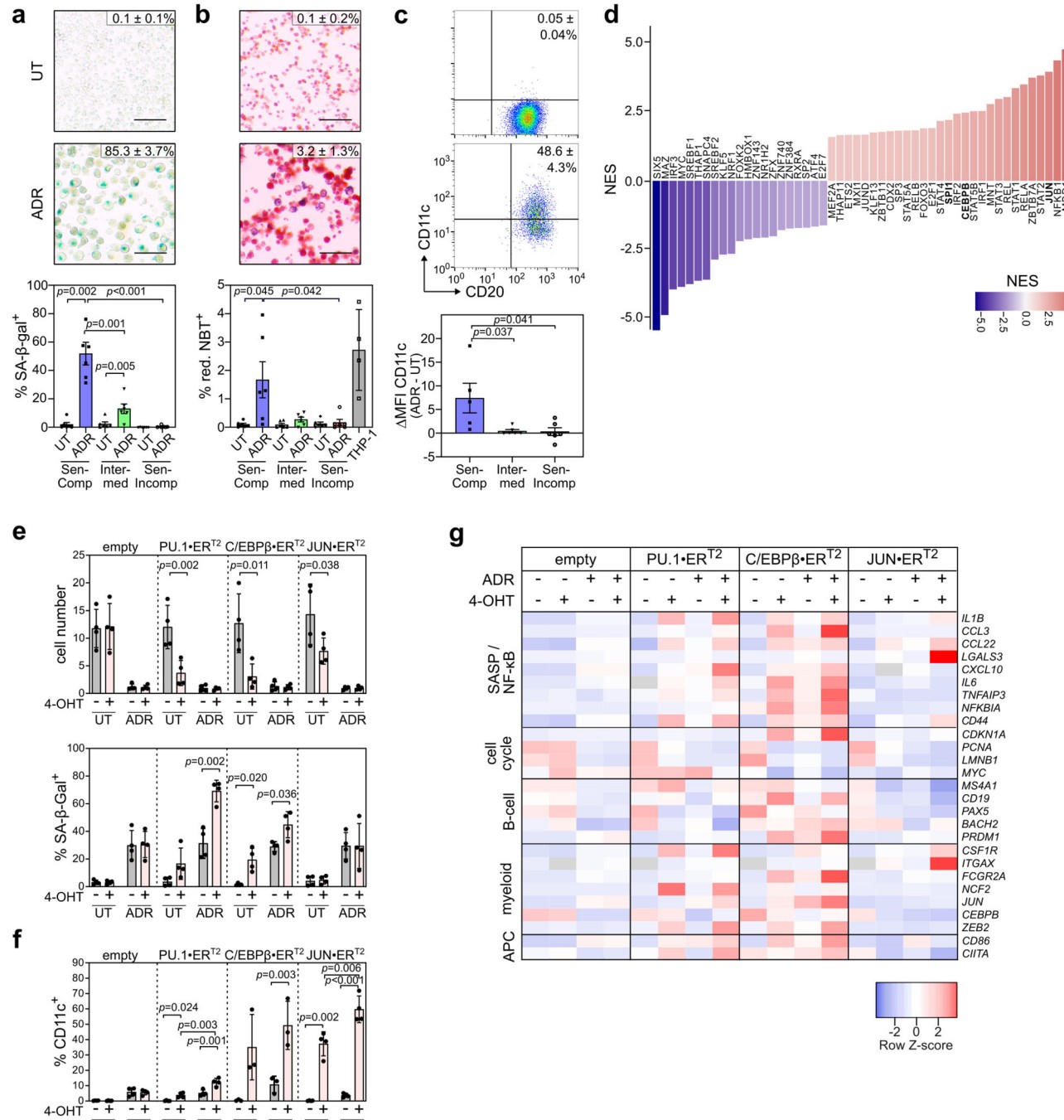

**Fig. 4 | TIS-associated gain of aberrant myeloid features in DLBCL cell line models. a, b** Senescence-associated β-galactosidase (SA-β-gal) and nitroblue-tetrazolium (NBT) respiratory burst staining of SU-DHL10 cells (senescence-competent) after 5-day adriamycin (ADR) exposure or left untreated (UT) (top; mean percentage of positive cells ± SD from *n* = 3 independent experiments). Scale bars, 100 μm. Quantification of SA-β-gal and NBT in DLBCL cell lines categorized as senescence-competent, -intermediate or -incompetent (bottom; *n* = 6 each). See Supplementary Data 4 for the group designation. **c** Mean percentage ± SD of surface CD11c⁺CD20⁺ on SU-DHL10 cells as in **a**, **b** (top). CD11c mean fluorescence intensity (MFI) difference between ADR-treated and UT cells across cell lines as in **a** (bottom). **a**–**c** Bars show mean percentage of positive cells ± SEM (**a**, **b**) or ΔMFI ± SEM (**c**), except for acute monocytic leukemia cell line THP-1 as positive control (**b**): ± SD (*n* = 4 independent measurements). Two-sided *P*-values by paired *t*-test for UT vs. ADR comparisons within same category, by unpaired *t*-test between categories. **d** DoRothEA TF footprinting of DLBCL cell line RNA-seq comparing 5-day ADR-treated, cell-cycle arrest-enriched to UT, proliferation-enriched cells.

Top 50 TF shifting activity comparing senescence-competent to -incompetent groups (*n* = 6 cell lines each) with normalized enrichment scores (NES) reflecting treatment-induced relative gain (red) or loss (blue) of TF activity. **e** Proliferation of Karpas-422 cells expressing PU.1-ER^T2, C/EBPβ-ER^T2, JUN-ER^T2 or empty vector, and 5-day exposed to ADR and/or 4-hydroxytamoxifen (4-OHT), or solvent (UT). Cell numbers as mean fold-change ± SD (top) and SA-β-gal in matching samples as mean percentage of positive cells ± SD (bottom). *n* = 4 independent experiments. **f** Percentage of CD11c⁺ cells by flow cytometry of cells as in **e**, except *n* = 3 for C/EBPβ-ER^T2. **e, f** *P*-values by two-sided, paired *t*-test. **g** Heatmap of qRT-PCR-based relative transcript abundance of cells as in (**e**). Transcripts are categorized as indicated; senescence-associated secretory phenotype (SASP), antigen-presenting cell (APC). Row *Z*-scores of fold-change geometric means compared to untreated, empty vector from *n* = 3 independent experiments. Gray indicates transcript levels below detection. *IL1B*, *CCL22*, *IL6*, and *CD44* fold-changes were log₂ transformed prior to *Z*-scoring. Source data are provided in the Source Data file.

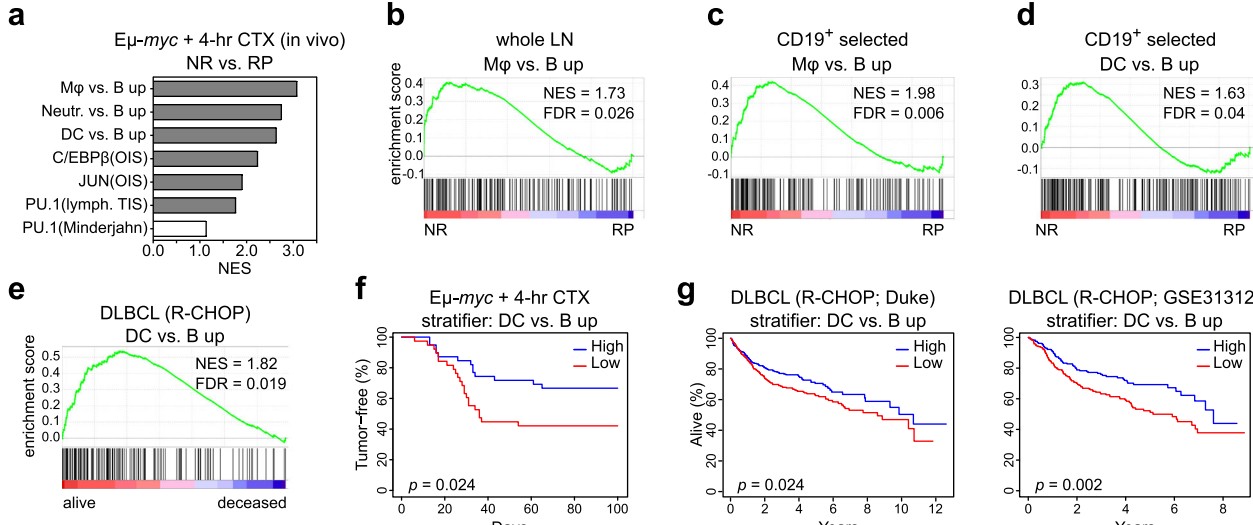

**Fig. 5 | TIS-associated myeloid plasticity underlies superior long-term outcome in DLBCL patients. a** Gene set enrichment analysis (GSEA) probing whole lymph-node (LN) lysate gene expression profiles (GEP) from newly diagnosed Eµ-*myc* lymphomas (GSE134751[39]) with the perspective of cure (i.e. never-relapse [NR]; *n* = 20) vs. relapse (i.e. relapse-prone [RP]; *n* = 19) that were 4-hr in vivo cyclophosphamide (CTX)-challenged. Gene sets distinguishing murine myeloid cell types from B-cells and of associated TF targets are analyzed. Positive normalized enrichment scores (NES) signify enrichment in the NR + 4-hr CTX group. Enrichments with false discovery rate (FDR) < 0.05 are marked gray. **b** GSEA of the 'macrophage (Mφ) vs. B-cell up' gene set for lymphomas as in **a**, but as treatment-naïve at baseline. **c** Specimens as in **b**, but selected for CD19+ B-cells, (NR and RP; *n* = 16 each). **d** As in **c**, but probing a murine gene set that distinguishes dendritic cells (DC) from B-cells. **e** GSEA probing a human DC vs. B-cell signature in a data set of human diffuse large B-cell lymphoma (DLBCL) patients[68] that remained event-free (alive) or not (deceased). *n* = 513 vs. *n* = 238 patients, respectively. **b–e** Enrichment plots with their respective NES and FDR values. **f** Kaplan-Meier plot of tumor-free survival observed in Eµ-*myc* lymphoma-bearing mice after a single-dose of CTX, stratified by average expression levels of the top 100 'DC vs. B up' genes in GEP of lymphoma LN 4-hr post in vivo CTX challenge as in (**a**). **g** Overall survival (OS) of rituximab, CTX, doxorubicin, vincristine and prednisone (R-CHOP)-treated DLBCL patients from two independent cohorts stratified by average expression levels of the human 'DC vs. B up' gene signature: the Duke cohort[68] (*n* = 751, left) and The International DLBCL Rituximab-CHOP Consortium Program (GSE31312; *n* = 470; right). **f**, **g** Survival curves reflect a median split and log-rank-based *P* values are shown. Source data are provided in the Source Data file.

survival of DLBCL patients (Supplementary Fig. 7d). Thus, the loss of MHC class I expression as a potential immune evasion strategy would not explain the survival advantage observed in the DC^high subgroup. Consistent with PU.1 being a key driver of DC-like plasticity[70], we also found a significantly better treatment outcome in mouse and human subsets with above-median-expression of the *Spi1* transcript (Supplementary Fig. 7e, f). Likewise, the humanized version of a TIS-specific PU.1 signature also marked DLBCL patients with longer overall survival (Supplementary Fig. 7g, Supplementary Data 5). Taken together, TIS-related PU.1 and DC signatures predict favorable long-term outcome in mice and humans with aggressive large B-cell lymphomas.

## Association of DLBCL cellular origin, genomics, states and eco-types with myeloid skewing

We next asked whether senescence-related aberrant plasticity might be associated with previously reported, molecularly defined DLBCL subsets. No obvious differences in their DC or PU.1 TIS signature scores were unveiled when comparing the otherwise R-CHOP-outcome-relevant GCB and activated B-cell (ABC) cell-of-origin (COO) subtypes[71,72] (Supplementary Fig. 8a). Thus, we interrogated the recently presented, more refined and heterogeneity-reflective prob-abilistic genomic subgroups (e.g. the 'LymphGen' algorithm[73]) as well as novel lymphoma ecotypes defined by distinct B-cell states (Bcs) and TME cell states (TMEcs) that were derived from multiple cell type-specific scRNA-seq training data sets and CIBERSORTx-deconvoluted bulk RNA-seq information[68,74–76]. DC and PU.1 gene signatures were prominent in BN2, ST2, and N1 LymphGen subtypes (Supplementary Fig. 8b), which are rather diverse by their COO relationship and asso-ciated prognoses[73]. Among the five Bcs reported, we found our DC- and monocyte/Mφ-skewed signatures to be most closely related to GC B-cell-associated conditions with pre-memory B-cell characteristics

(Bcs S2, and, if slightly more differentiated, Bcs S3; Supplementary Fig. 8c). Out of the Bcs/TMEcs-integrative classification of nine DLBCL lymphoma ecotypes (LE) ordered in association with patient overall survival from shortest (LE1) to longest (LE9)[76], DC and PU.1 gene sig-natures scored highest in LE 4, 7 and 9 (Fig. 6a, Supplementary Fig. 8c). Notably, a river plot of Eµ-*myc* lymphoma transcriptomes assigned to best-fit LE prior to therapy compared to their 4-hr CTX-treated mat-ches showed the expected heterogeneity among the LE distributions at baseline, and unveiled a significant dynamic shift of drug-exposed lymphomas towards the TIS-related DC/PU.1 gene signature-enriched LE7 and an emerging LE9 group, hence, convergence towards the best-outcome-associated LE groups (Fig. 6b). Collectively, these results suggest aberrant plasticity as a senescence-related modifier of the cell-autonomous lymphoma biology and its related ecostate, thereby adding an important independent and dynamic layer to the hitherto rather static genome- and transcriptome-based classifiers of the typi-cally bulk-analyzed DLBCL samples.

## The DC^high status marks a favorable MyD88-mutant DLBCL subgroup

Although the prognostically rather poor MCD subtype, which com-prises the majority of MyD88-mutant cases (specifically those with the most prevalent L265P mutation)[75], displayed limited association with DC/PU.1 signatures (Supplementary Fig. 8b), we decided to more deeply investigate lymphomas with mutant MyD88, because this toll-like receptor adaptor molecule reportedly operates upstream of AP-1/JUN[77]. Moreover, we previously linked MyD88 mutations to macro-phage attraction, profound senescence induction, T-cell recognition, and immune evasion in murine and human aggressive B-cell lymphoma[6]. We reanalyzed RNA-seq data generated from B-cell-purified lymphomas that formed on an Eµ-*myc*;MyD88-L265P

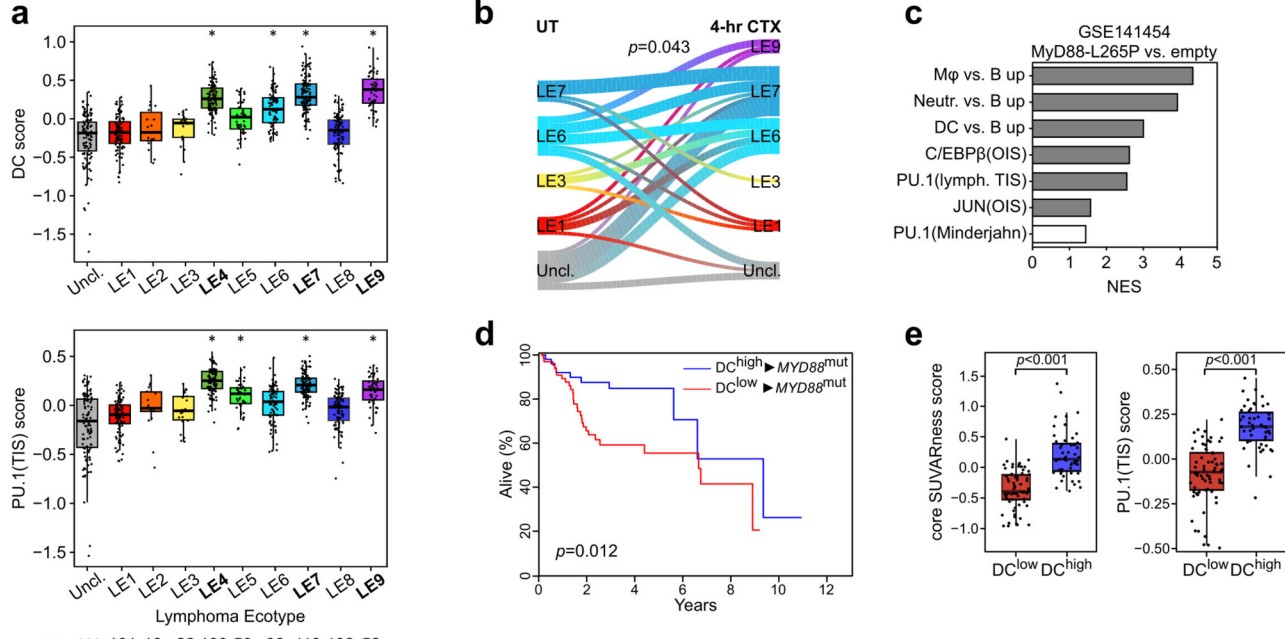

**Fig. 6 | Senescence-associated myeloid skewing links novel DLBCL subtypes to favorable outcomes. a** Average expression scores of "dendritic cell (DC) vs. B cell up" (DC score, top) and "PU.1 targets in TIS" (PU.1[TIS] score, bottom [cf. Supplementary Data 5]) signatures in subgroups of the Duke diffuse large B-cell lymphoma (DLBCL) cohort[68] classified into the indicated lymphoma ecotypes (LE; n as indicated in the figure)[76]. * marks ecotypes with significantly higher scores than at least half of the other LE groups (Benjamini-Hochberg adjusted $P < 0.05$, see Source Data for the exact $P$-values for each pairwise comparison), as assessed by pairwise two-sided Wilcoxon rank-sum test; LEs marked high for both gene signatures are depicted in bold. "Uncl." denotes unclassified cases. **b** River plot showing dynamic changes of the gene expression profile (GEP)-based LE designation between matched pairs of untreated (UT) and 4-hr cyclophosphamide (CTX)-in vivo-challenged Eμ-myc lymphomas (GSE134751; n = 37 each). Only LE classifications with at least three samples were considered (hence, LE4 was excluded). Pearson's χ² P value is

displayed. **c** Gene set enrichment analysis comparing GEP of lymphomas that formed from Eμ-myc transgenic hematopoietic stem cells with or without stable introduction of a MyD88-L265P lesion in recipient mice (n = 7 each; GSE141454[6]). Positive normalized enrichment scores (NES) signify enrichment in MyD88-L265P+ lymphomas. Gene sets scoring with false discovery rate (FDR) < 0.05 are marked gray. **d** Kaplan-Meier plot of overall survival and log-rank $P$ value in the Duke cohort stratified as in Fig. 5g, but selected for patients with a MyD88 mutation (DC^high, n = 51 vs. DC^low, n = 66). **e** GEP of DLBCL patients as in **d**, scored for the average expression of the core SUVARness senescence signature[39] (left), and the PU.1(TIS) signature (right). $P$-values by two-sided Wilcoxon rank-sum test. **a, e** Box plots show the median as center line, 25% and 75% percentiles as box limits, and whiskers extending to largest and smallest values within the 1.5× interquartile range of the box limits. Source data are provided in the Source Data file.

---

background, and found a strong enrichment for myeloid signatures in the MyD88-L265P vs. Myc-only driven lymphomas (Fig. 6c). Moreover, MyD88-L265P-driven lymphomas enriched for PU.1 and C/EBPβ gene sets, and exhibited high-level expression of the respective TF and their target genes, such as *Csf1r* and *Il1b* (Fig. 6c, Supplementary Fig. 8d). We, therefore, hypothesized that senescence-prone MyD88-mutant B-cell lymphomas might represent a subgroup with intrinsic similarities to the TIS-associated aberrant phenotype switch. Focusing on MyD88-mutant cases in DC-stratified DLBCL patients[68], we found the DC^high/MyD88^mut subgroup to experience a significantly better survival than the DC^low/MyD88^mut subgroup (Fig. 6d). Furthermore, we detected a substantially higher expression of our previously established senescence-indicative SUVARness signature[39] (Fig. 6e, left), and of the PU.1 TIS signature (Fig. 6e, right). Lastly, while DC^low/MyD88^mut DLBCL cases are strongly enriched for in the poorest outcome LE, i.e. LE1, DC^high/MyD88^mut are predominantly found in LE4, LE6, LE7 and LE9, indicating a clear shift to superior long-term outcome (Supplementary Fig. 8e). These findings underscore that TIS-related aberrant plasticity marks a favorable subset within the MyD88-mutant DBLCL subgroup with its globally rather poor outcome.

## PU.1 links antigen presentation and immune checkpoint control to TIS

To mark candidate genes that became accessible to PU.1 by senescence-associated chromatin remodeling, we performed pathway annotation of transcripts that were positively regulated by PU.1 in TIS

but not in non-senescent murine Eμ-myc;bcl2 lymphomas (Supplementary Data 5). Applying an InnateDB pathway overrepresentation analysis[78] using KEGG (Kyoto encyclopedia of genes and genomes) and REACTOME databases, we found, among other immune terms, 'interferon gamma signaling' as well as the 'antigen processing and presentation' pathway, containing structural MHC class II (*H2-Ab1*, *H2-DMb1*, *H2-Eb1*) and MHC II regulator transcripts (*Ciita*, *H2-DMa* and *Ctss*), strongly enriched (Fig. 7a, Supplementary Data 5, cf. Figure 3c, e). This was further validated by reduced upregulation of surface MHC II and of the costimulatory CD86 molecule on PU.1-depleted TIS compared to matched lymphoma cells with a control shRNA (Fig. 7b; see Supplementary Fig. 9a for no significant impact on MHC I). Since we recently showed profoundly enhanced surface expression of immune checkpoint ligands PD-L1 (encoded by *Cd274*) and PD-L2 (encoded by *Pdcd1lg2*) on lymphoma cells in an OIS-dependent fashion[6], we sought to examine here PD-L1 expression in the DC-like mouse lymphoma TIS subpopulations. Indeed, CD11b+ and CD115+ cells displayed significantly higher levels of cell surface PD-L1 in TIS (Fig. 7c), with much higher PD-L1/2 transcript inductions found in CD115+ compared to CD115− TIS lymphoma cells (Fig. 7d). We identified PU.1 as the driving TF behind, because TIS-related high-level PD-L1 expression was markedly attenuated upon *Spi1* knockdown (Fig. 7e). Consistent with no PU.1 induction detectable upon ADR treatment of senescence-incapable Suv39h1-deficient lymphoma cells (see above), MHC I and II, CD86 and PD-L1 expression levels remained largely unchanged or were much less induced compared to Suv39h1-proficient cells

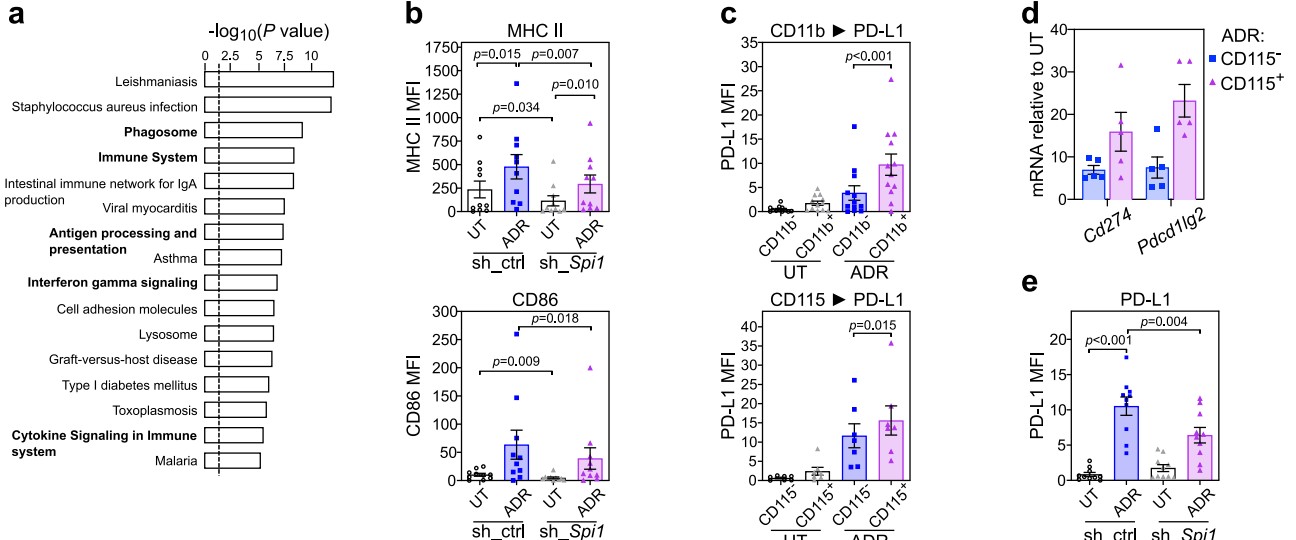

**Fig. 7 | TIS-associated PU.1 drives antigen presentation and expression of immune checkpoint molecules. a** InnateDB KEGG and REACTOME pathway overrepresentation analysis[78] of therapy induced senescence (TIS)-associated PU.1 targets. Hypergeometric distribution test, Benjamini-Hochberg multiple testing corrected *P* values for each pathway are shown. See Supplementary Data 5 for scored transcripts. Terms relevant for tumor immunity in bold. **b** Flow cytometry analysis of MHC class II (top) and CD86 (bottom) expression on untreated (UT) and 5-day adriamycin (ADR)-exposed control;*bcl2* lymphomas, transduced with a *Spi1* targeting (sh_*Spi1*) or control shRNA (sh_ctrl) (*n* = 10 individual lymphomas each).

**c** Flow cytometry analysis of PD-L1 (CD274) expression on individual UT and 5-day ADR-exposed control;*bcl2* lymphomas, gated by CD11b (*n* = 12, top) or CD115 (*n* = 7, bottom) expression. **d** qRT-PCR-determined expression of *Cd274* and *Pdcd1lg2* (encoding PD-L2) in individual ADR-exposed control;*bcl2* lymphomas, FACS-sorted by CD115 expression (*n* = 5; samples as in Fig. 2c). Bars represent the mean fold-change relative to matched UT lymphomas ± SEM. **e** As in **b**, but analyzing surface PD-L1 expression. **b, c, e** Data points are isotype-corrected MFI values and bars represent the average mean fluorescence intensity (MFI) ± SEM. *P*-values by two-sided, paired *t*-test. Source data are provided in the Source Data file.

(Supplementary Fig. 9b). We also tested whether PU.1 induction might drive expression of those immune modulators, and found elevated CD86 and PD-L1 expression in human Karpas-422 DLBCL cells upon 4-OHT-enforced PU.1 expression in the absence of ADR, thereby positioning PU.1 downstream or independent of senescence-associated priming of these target gene promoters (Supplementary Fig. 9c). Our data mark PU.1 as a key regulator of immunostimulatory and -inhibitory molecules and promoter of a lineage-aberrant DC-like phenotype in TIS with functional reminiscence to professional antigen-presenting cells (APC)[79], thereby pinpointing its role as critical rheostat of T-cell mediated immune control of cross-differentiated senescent lymphomas.

## TIS-related DC-like lymphoma reprogramming underlies T-cell immunosurveillance

Because cardinal DC functions are antigen presentation and T-cell activation, thereby potentially contributing to superior outcome of DC-skewed lymphomas, we postulated lineage infidelity upon senescence induction to prime for T-cell recognition and eventual T-cell-mediated tumor control. To test and characterize the immunogenicity of TIS cells, we inoculated strain-matched mice with ex vivo-pretreated (i.e. TIS) or untreated Eμ-*myc;bcl2* lymphomas. Two weeks later, we isolated from spleens thus senescence- or control-primed T-cells ($T_{TIS}$ vs. $T_{UT}$, respectively), or lymphoma-naïve T-cells of tumor-free mice ($T_{naïve}$) for comparison (Fig. 8a). We then co-cultured these three T-cell types with the same individual lymphomas used for $T_{TIS}$ or $T_{UT}$ generation, provided as either untreated or in TIS, ex vivo. Notably, TIS lymphomas showed the strongest loss in viability (tested after 72 h in co-culture) when exposed to $T_{TIS}$, and were also highly susceptible to lymphoma-educated $T_{UT}$, while effects were much weaker if the target lymphoma was not pre-treated into TIS, thereby underscoring the enhanced immunogenicity of senescent lymphomas and the superior recognition by TIS lymphoma-primed T-cells (Fig. 8b). Accordingly, TIS lymphomas induced significantly higher cytotoxic

T-cell activation levels, as evidenced by the highest fraction of CD69+ cells in the CD8+ subpopulations of $T_{TIS}$ when compared to $T_{UT}$ or $T_{naïve}$ cells, and with activation levels consistently higher compared to those elicited by untreated lymphomas (Fig. 8c). Hence, the TIS state switch elicits an immunogenic mechanism that renders senescent cells particularly attractive targets especially for TIS-educated T-cells.

Next, we sought to explore the MHC dependency of $T_{TIS}$-mediated cytotoxicity in greater detail, for which we separated individual $T_{TIS}$ specimens into CD4+ and CD8+ T-cells. While both CD4+ and CD8+ T-cells contributed to TIS lymphoma elimination, we found CD4+ T-cells significantly more potent than CD8+ T-cells (Fig. 8d). Accordingly, blocking MHC II partially rescued TIS cells from T-cell-mediated killing (Supplementary Fig. 10a). Of note, since MHC II ligation on its own resulted in some cytotoxicity[80], the actual contribution of TIS-associated class II antigen presentation for T-cell-mediated cytotoxicity might be even greater.

To test if lineage-aberrant plasticity contributes to the immunogenicity of TIS cells, we interrogated lymphoma cells upon knockdown of PU.1 and found them to be significantly protected against $T_{TIS}$-mediated killing, paralleled by significantly reduced activation of CD8+ $T_{TIS}$ cells (Fig. 8e, f). This is in line with PU.1-driven MHC II expression and subsequent activation of CD4+ T-cells, which seem to contribute as cytotoxic effector and CD8+ T-cell-stimulating helper cells, because CD8+ T-cell activation was found reduced upon MHC II blockade (Supplementary Fig. 10b). Moreover, immune checkpoint blockade (ICB) using an anti-PD-L1 antibody further enhanced the cytotoxic attack via now de-repressed $T_{TIS}$ cells at their TIS lymphoma targets, while such effect was not seen with $T_{UT}$ cells under ICB, or when PU.1 expression was reduced, indicating a specific adaptive T-cell response component underlying DC-like TIS lymphoma cell killing (Fig. 8g, h).

Conversely, the DC-like transition via enforced PU.1 or C/EBPβ expression, promoting antigen presentation in TIS as measured by a fluorogenic ovalbumin substrate (Supplementary Fig. 10c), rendered TIS-capable but not Suv39h1-deficient Eμ-*myc;bcl2* lymphomas more

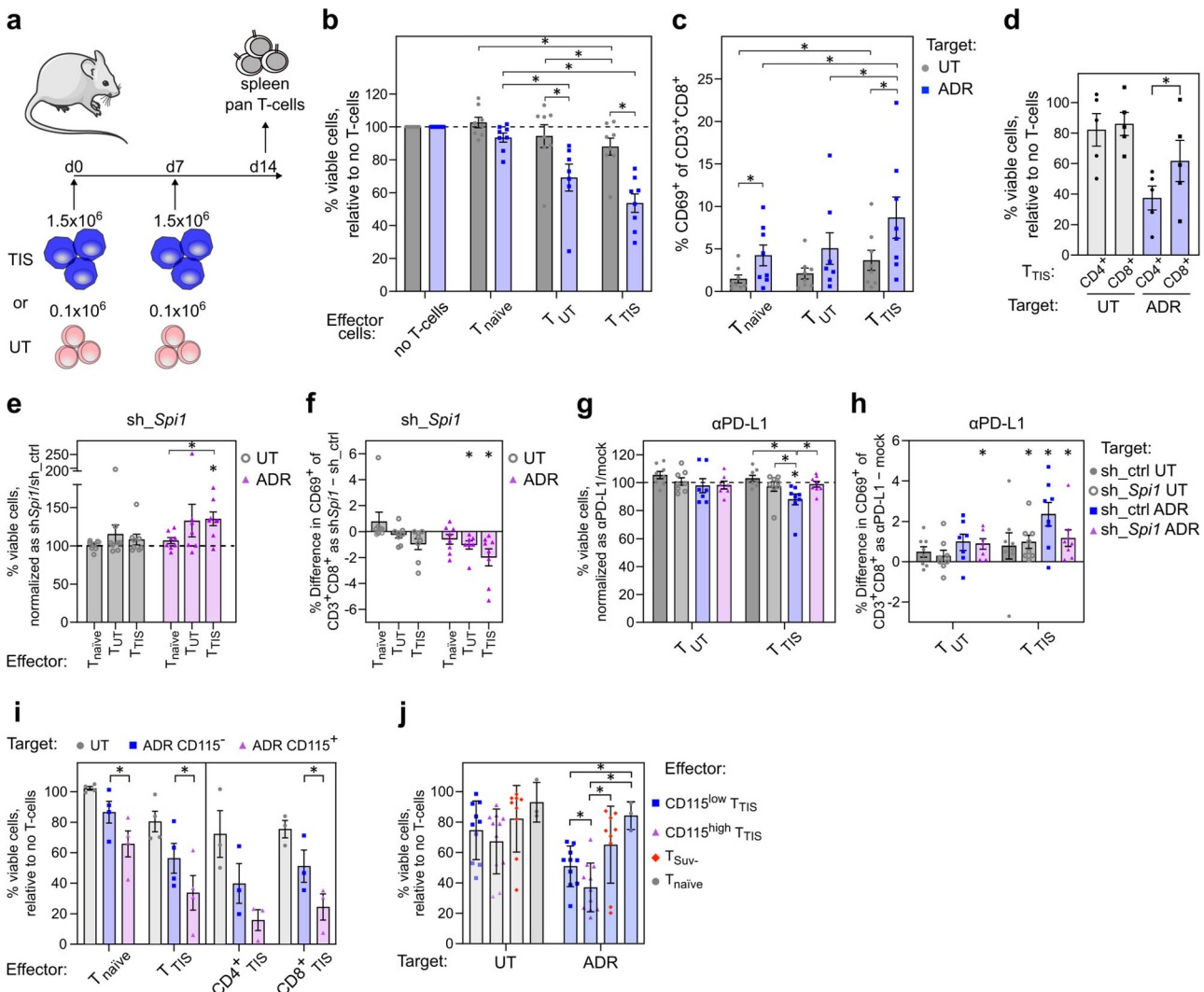

**Fig. 8 | DC-like reprogramming promotes TIS-preferential T-cell-governed anti-lymphoma immunosurveillance. a** Mouse immunization by inoculating therapy-induced senescent (TIS) or treatment-naïve (UT) control;*bcl2* lymphomas twice. Primed spleen pan-T-cells isolated two weeks later. **b** Viability of control;*bcl2*;sh_ctrl lymphomas 6-day-exposed to adriamycin (ADR) or left untreated (Target; *n* = 8 lymphomas), then co-cultured for 72-hr with T-cells (Effector) from **a** ($T_{TIS}$ or $T_{UT}$) or from wild-type mice ($T_{naïve}$). **c** Activation of CD3$^+$CD8$^+$ T-cells as mean percentage CD69$^+$ ± SEM after co-culture from **b**. **d** Viability of individual UT vs. ADR-exposed control;*bcl2* lymphomas (*n* = 5 each) 48-hr co-cultured with CD4$^+$ and CD8$^+$ $T_{TIS}$ subsets. **e** As in **b**, but for *Spi1*-depleted lymphomas (sh_*Spi*). Viabilities normalized to matched conditions of paired sh_ctrl lymphomas (**b**). **f** As in **c**, but showing the difference in CD69$^+$CD8$^+$ T-cells between matched sh_*Spi1* and sh_ctrl specimens. **g** Analogous to **b** and **e**, but assessing PD-L1 blockade effect (αPD-L1) on lymphoma viability (legend on the right) post co-culture with $T_{TIS}$ or $T_{UT}$. Viabilities normalized to equivalent co-cultures without αPD-L1. **h** Analogous to **c** and

**f**, showing CD69$^+$ differences on CD8$^+$ T-cells between αPD-L1- and mock-treated co-cultures. **i** Viability of TIS lymphomas sorted into CD115$^+$ and CD115$^-$ fractions before co-culture with $T_{TIS}$ cells primed with unsorted, matched TIS lymphomas, or with $T_{naïve}$ cells (*n* = 4 for $T_{TIS}$ and $T_{naïve}$, or *n* = 3 for CD4$^+$- and CD8$^+$-$T_{TIS}$). Bars represent relative mean percentage of viable cells ± SEM compared to respective cell cultures without T-cells (**b, d, e, g, i**) or mean percentage difference of CD69$^+$CD3$^+$CD8$^+$ cells ± SEM (**f, h**). **j** Viabilities of individual lymphomas as in **b** (*n* = 3) after 48-hr in non-matched co-cultures, i.e. target lymphomas were different from the $T_{TIS}$-priming lymphoma, with CD115$^{low}$ $T_{TIS}$ (*n* = 4), CD115$^{high}$ $T_{TIS}$ (*n* = 4; cf. Supplementary. Fig. 10f), $T_{Suv}$ (*n* = 3; splenic T-cells primed by chemotherapy-exposed *Suv39h1*$^-$;*bcl2* lymphomas), or $T_{naïve}$ (*n* = 1). Bars show mean percentages ± SD. * two-sided *P* < 0.05 by paired (connecting lines) or one-sample *t*-test (**b-i**) or by unpaired *t*-test (**j**). See Supplementary Fig. 11a, b for combined data panels of **b, c, e-h**. See source Data file for exact *P*-values.

susceptible to $T_{TIS}$ and/or $T_{naïve}$ killing, and resulted in higher cytotoxicity against lymphomas in TIS (Supplementary Fig. 10d). Notably, while enforced PU.1 induction without additional ADR exposure promoted the B-to-M transition as indicated by expanding fractions of CD115$^+$ cells in TIS-capable control;*bcl2* cells, no such effect was seen in TIS-incapable Suv39h1-deficient cells, presumably reflecting the absent cooperation of senescence-associated chromatin remodeling with PU.1 action towards myeloid cross-differentiation in the latter, thereby explaining the observed lack of ADR-inducible T-cell immunogenicity in these TIS-incapable lymphomas (Supplementary Fig. 10e).

In line with PU.1's role in TIS-related myeloid skewing, we found CD115$^+$ TIS cells to be much more susceptible to $T_{TIS}$- and even $T_{naïve}$-mediated lysis than their matched CD115$^-$ counterparts, underscoring the enhanced propensity of lineage-aberrant, DC-like TIS cells to become targets of both CD4$^+$ and CD8$^+$ T-cell recognition (Fig. 8i). Importantly, TIS lymphomas which were used to prime T-cells in vivo exhibited much higher expression of the costimulatory molecule CD86 if they were CD115$^{high}$ (Supplementary Fig. 10f). Specifically, we tested the ability of different CD115$^{high}$ vs. CD115$^{low}$ lymphoma-primed $T_{TIS}$ isolates to lyse three different individual target lymphomas (but excluding in this analysis the matching lymphoma/T-cell constellations

used for in vivo-priming before). Importantly, $T_{TIS}$ cells primed by CD115$^{high}$ lymphomas showed significantly stronger TIS-eliminating capacity when compared to CD115$^{low}$-primed $T_{TIS}$ cells, with the latter being still far more efficient than $T_{naïve}$ cells (Fig. 8j). Notably, limited T-cell cytotoxicity was elicited by T-cells primed by drug-exposed Suv39h1-deficient lymphoma cells, underscoring the dependency of efficient T-cell priming specifically on the senescent and DC-like state switch, not mere chemotherapy exposure. Hence, myeloid-skewed TIS lymphomas provide stronger immunogenic signals, activate T-cells better, and present themselves as superior targets for T-cell-mediated senolysis as evidenced by reduced sensitivity of PU.1-depleted lymphomas and by higher susceptibility of the CD115$^+$ TIS subpopulation.

Collectively, these findings support the notion that a subset of DLBCL cases within the heterogeneous biological spectrum of this entity possesses a senescence-linked cell-intrinsic capability to aberrantly differentiate across lineage borders, and, more specifically, to adopt DC-like functionalities, which may result in enhanced immune surveillance and improved outcome to R-CHOP-based standard-of-care induction therapy. Our results further suggest an ICB extension of R-CHOP as a testable first-line concept for DC-high DLBCL patients with additional risk features.

## Discussion

Cellular senescence, often reduced to a terminal cell-cycle arrest associated with a pro-inflammatory secretome, reflects a condition of fundamentally reprogrammed cell functions due to profound epigenetic and transcriptional remodeling. Adding to our previously reported finding on senescence-associated stemness[16], we present here aberrant plasticity as a hitherto underappreciated layer of the multi-faceted senescent state switch and the key mechanism underlying the just emerging role of T-cell recognition in senescence[6,81–85]. With certain similarities to the enforced B-to-M conversion by a stepwise TF-based reprogramming protocol[34], and reminiscent of a myeloid bias reported in aged hematopoietic stem cells with features of replicative senescence[86–88], we dissected here the senescence-remodeled AP-1-, C/EBPβ- and PU.1-governed TF network as a driver of aberrant and immunogenic lineage-crossing differentiation from malignant B-cells to Mφ/DC-like cells. The functional ramifications are far-reaching, since these aberrantly B-to-M-transitioned cells not only adopt a myeloid GEP with variable preservation of their B-cell phenotype but gain phagocyte-reminiscent oxidative burst capacity, much more effectively activate T-cells, and promote an adaptive PD-L1/2:PD1-sensitive MHC/TCR-mediated immune response. Adding to previous findings by us and others[9,15,16], we view lineage-promiscuous myeloid plasticity next to SASP production and senescence-associated stemness as another physiologic capability by which senescent cells contribute to the dynamic process of proper wound healing: upon tissue injury, primarily damaged and their co-affected surrounding cells enter senescence, remodel extracellular matrix, attract immune cells, remove debris, replenish parenchymal cellularity and eventually get cleared once stresses no longer apply[1].

Immune-mediated elimination of senescent cells operates as an important principle in tissue homeostasis to terminate less desirable cell-autonomous and environmental effects lastingly persistent senescent cells might otherwise account for. Our results pinpoint T-cell priming by plastic senescent cells as a critical component of the self-limiting life-cycle of senescent cells[1,7–9], and address the specific implications of such principle in a malignant context. Functional ex vivo-assays demonstrated the collaborative interaction between DC-like cross-differentiated B-lymphoma cells and T-cells, and underscored the prognostically favorable role of such myeloid skewing for enhanced tumor immunosurveillance in long-term observed murine and human lymphoma cohorts in vivo. Despite the superior tumor control conferred by DC-like tumor cell plasticity, we found TIS-associated activation of immune checkpoints to potentially blunt

tumor-directed T-cell cytotoxicity, thereby suggesting ICB as an exploitable co-treatment strategy. The underlying contributions of an MHC-presented, antigen-dependent and TCR-governed T-cell control on one hand and enhanced SASP-driven, inflammation-like TCR-independent bystander activation leading to, for instance, Fas-mediated killing on the other hand require further in-depth investigations[89,90].

The lymphoma micromilieu, beyond the mere malignant cell population, is increasingly being recognized as fate-decisive: distinct DC and Mφ states within recently inaugurated DLBCL ecotypes were linked to differences in long-term responsiveness[76]. Notably, DC or Mφ were part of most LE identified – spanning LE associated with the worst and the best outcomes to standard R-CHOP-like therapy. While aggressive B-lymphoma cells were shown to coopt and educate adjacent normal DC to provide tumor-supportive pro-survival signals reminiscent of the detrimental LE1 characteristics[91,92], we elucidated here a very different kind of senescence-associated 'neo-plastic' micromilieu that mimics bystander Mφ and DC but actually originates cell-autonomously from cross-differentiated tumor cells and boosts the lymphoma-controlling interaction with host T-cells. Biologically, the so-composed lymphoma environments are more similar to human DLBCL LE with good to excellent prognosis, relate to distinct Mφ or DC states compared to Mφ or DC in poorer prognosis LE, and may dynamically shift from one LE to another, with the senescent state switch typically reflected by the transition to a better-prognosis LE such as LE7 and LE9[76]. Beyond the cellular immune interactions we elucidated here, it will be interesting to investigate small compounds that exploit cross-differentiation-associated vulnerabilities to directly target cancer cells, similar to a strategy we pursued to pharmacologically resensitize lineage-promiscuous Hodgkin's lymphoma cells to B-cell-typical targeting therapeutics[33].

In essence, we mechanistically dissected here senescence-related remodeling of a lineage-controlling transcription network consisting of AP-1, C/EBPβ, and PU.1 TF in murine and human aggressive B-cell lymphomas. Mimicking Mφ- and DC-rich environments by neoplastic cell-autonomous B-to-M cross-differentiation, deregulation of these master TF had far-reaching implications for the lymphoma ecostate and its T-cell-governed immunosurveillance. Moreover, TCR-dependent and -independent modes of T-cell activation evoked T-cell-mediated cytotoxicity not only against lineage-aberrant lymphoma cells but also, albeit to a lesser extent, lineage-non-promiscuous lymphoma cells. While explaining, at least in part, the superior long-term outcome of patients whose tumors exhibit such beneficial senescence-associated immune-biology, the findings ask for careful examination of senolytic agents in this context – when to apply, and whether complete clearance of immunogenic, T-cell memory-maintaining senescence-arrested persister cells would be most beneficial. Our observations have important ramifications for diagnostic profiling and treatment-induced dynamics of tumor ecostates, and aberrant plasticity-guided targeting strategies.

## Methods

### Mouse strains and lymphoma generation

All animal protocols used in this study were approved by the governmental review board (Landesamt Berlin) and conform to the appropriate regulatory standards. All mice were in a C57BL/6 strain background and included both sexes. Eµ-myc transgenic lymphomas with defined genetic defects in the Suv39h1 locus were generated by cross-breeding to Suv39h1 knockout mice[36,38]. Genotyping of the offspring by allele-specific genomic PCR, monitoring of lymphoma onset and isolation of viable lymphoma cells were carried out as described[69,93]. C57BL/6 N mice for in vivo lymphoma propagation were bought from Charles Rivers. Animals were housed under specific-pathogen-free conditions in individually ventilated cages and a 12 h light/dark cycle, 20–22 °C, and 50–60 % relative humidity. The maximum tumor burden for lymphoma-bearing mice of 16 mm as

measured by the total diameter of enlarged lymph nodes was not exceeded. Termination criteria included weight loss of >20 %, marked ascites, paralyses, and dyspnea.

## Plasmids

Bcl2 and Hras[V12] overexpression plasmids have been described[17,36,38,54]. To achieve global AP-1 repression, a hemagglutinin (HA) tagged cJunΔN moiety was generated by PCR amplification of the mouse *Jun* transcript, using custom primers resulting in specific deletion of the first N-terminal 122 amino acids. The resulting product was then sub-cloned into MSCV-IRES-GFP and MSCV-PGK-DsRed backbones. cDNAs encoding mouse *Spi1* (PCR-amplified from mouse lymphoma cDNA), rat *Cebpb* and human *CEBPB* (amplified from MSCV-rCEBPB-ER-IRES-GFP and pcDNA3-LAP*, respectively, both kind gifts from Achim Leutz), human *SPI1* (amplified from Karpas-422 cDNA), and human *JUN* (amplified from PMA treated THP-1 cDNA) were fused in frame with a (Z)-4-hydroxytamoxifen (4-OHT, Sigma-Aldrich) -inducible estrogen receptor mutant ER[T2] residing in MSCV-PGK-DsRed or MSCV-PGK-puro vectors[16]. All plasmid orf sequences were verified to encode wild-type protein sequence.

To achieve stable knockdown of mouse *Jun* and *Cebpb*, short hairpin RNAs (shRNA) were obtained in the shERWOOD pLMN vector from Transomic Technologies, along with a non-targeting shRNA-control vector. Mouse *Spi1* was targeted by a published shRNA sequence - (shPU.1_2 sense - 5′-GAAGCTCACCTACCAGTTC-3′)[94], inserted into the H1 promoter-driven pSuperRetro (Oligoengine) vector. Control oligonucleotides containing a scrambled *Spi1* sense sequence: 5′-GCCTAATCAACCGATTCCG-3′ were used to generate shRNA-control vectors.

## Cell culture and retroviral gene transfer

Eμ-*myc* lymphoma cells (LC) were cultured on irradiated NIH3T3 fibroblast feeders[95] in B-cell medium, and C57BL/6-derived ANA-1 macrophages were cultured in RPMI-1640 (Gibco) as previously reported[16,36]. Human diffuse large B-cell lymphoma (DLBCL) cell lines SU-DHL4, SU-DHL5, SU-DHL6, SU-DHL10, OCI-LY1, OCI-LY7, OCI-LY18, RI1, U2932, RC-K8, DB, WSU-DLCL2, CARNAVAL, DOHH2 and NU-DHL1 were purchased from DSMZ (Leibniz-Institut Deutsche Sammlung von Mikroorganismen und Zellkulturen GmbH), Toledo and SU-DHL2 were obtained from ATCC (American Type Culture Collection) and Karpas-422 from ECACC (European Collection of Authenticated Cell Cultures). DLBCL and THP-1[54] cell lines were cultured according to supplier's recommendations either in RPMI-1640 or IMDM media, supplemented with 10% FBS and 100 U ml⁻¹ penicillin-streptomycin, and all cell lines were regularly tested for mycoplasma contamination. Human diploid fibroblasts (HDF) were cultured as previously reported[54]. Retroviral supernatant for gene transfer experiments was generated by transient transfection of Phoenix packaging cells with murine stem cell retro-virus (MSCV)-based vectors[93]. Freshly isolated LC were stably trans-duced to express Bcl2[16]. Karpas-422, RC-K8, and SU-DHL5 were engineered to express the murine, ecotropic virus receptor *Slc7a1* (a.k.a. mCAT-1) as previously described[16]. HDF IMR90, Wi38, and Tig3 were transduced to express constitutive or 4-OHT-inducible onco-genic Hras-V12[17,54]. LC were further stably transduced with either pSR-sh*Spi1*, pLMN-sh*Jun*(ULTRA-3224229), pLMN-sh*Cebpb*(ULTRA-3208937), MSCV-cJunΔN-GFP or MSCV-cJunΔN-DsRed, MSCV-Spi1•ER[T2]-DsRed, MSCV-rCebpb•ER[T2]-DsRed or non-targeting shRNA in pLMN vector, scrambled shRNA in pSR or empty MSCV retroviruses, FACS sorted for fluorescent protein expression where appropriate and/or selected with either puromycin (5 μg ml⁻¹) or G418 (2 mg ml⁻¹). Human cell lines made susceptible to ecotropic infection were trans-duced with either MSCV-SPI1•ER[T2]-puro, MSCV-CEBPB•ER[T2]-puro, MSCV-JUN•ER[T2]-puro, or control MSCV-ER[T2]-puro retroviruses. Karpas-422 cells were selected with 1 μg ml⁻¹, RC-K8 cells with 2 μg ml⁻¹, and SU-DHL5 cells with 0.5 μg ml⁻¹ puromycin.

## In vitro-treatments, senescence induction

Mouse LC were seeded at $0.75\text{-}1 \times 10^6$ cells ml⁻¹ and treated once with 0.05 μg ml⁻¹ of doxorubicin (a.k.a. adriamycin, ADR), a topoisomerase II inhibitor, to induce senescence. After three days, the medium was exchanged to drug-free medium. DLBCL cell lines were seeded at $10^5$ cells ml⁻¹, except for SU-DHL5 ($2.0 \times 10^5$ cells ml⁻¹) and treated with sublethal concentrations of ADR (see Supplementary Data 4) for three days, typically resulting in a viability drop to 20-50% at day five after drug exposure. Mouse LC were gamma-irradiated with 8 Gy and left to senesce for seven days. Palbociclib (AbMole BioScience) was added to a concentration of 5 μM to mouse LC and maintained after interim medium exchange. ER[T2] fusion constructs were induced by adding 1 μM (murine cells) or 200 nM (human cells) 4-OHT. Cellular senescence readouts were performed five days after treatment unless stated otherwise.

## Analysis of cell viability, cell-cycle and senescence parameters

Cell viability and numbers were determined using Guava easyCyte HT12 or HT8 (Luminex) benchtop flow cytometers and the Guava ViaCount reagent (Luminex), or propidium iodide (PI; 1 μg ml⁻¹; Sigma-Aldrich) in conjunction with fluorescent protein-expressing speci-mens. Senescence was assessed by senescence-associated β-galacto-sidase (SA-β-gal) activity in freshly fixed cells at pH 5.5 (murine) or pH 6.0 (human cells) as previously described[16,36].

## Respiratory burst assay

Mouse or human cells were seeded in an appropriate medium supplied with 12-Phorbol myristate 13-acetate (PMA; 10 μg ml⁻¹; Sigma-Aldrich) and 20% FBS. After the addition of an equivalent volume of a Nitroblue tetrazolium chloride (NBT; 2 mg ml⁻¹; Roth) solution in PBS, cells were incubated at 37 °C, 5% $CO_2$ and atmospheric $O_2$ in a humidified incu-bator for 60 min. Samples were washed once with 1× PBS and fixed and counterstained with 0.1% Safranin O (Roth) in 100% ethanol for 5 min. Brightfield images were acquired under an EVOS AMEX-1200 microscope.

## DQ-OVA assay

Antigen uptake and processing assay was performed by incubating cells with DQ™ ovalbumin substrate (Thermo Fisher Scientific, DQ-OVA) at 10 μg ml⁻¹ for 2 h at 37 °C. Simultaneously, a control aliquot of cells was incubated with DQ-OVA at 4 °C to acquire background fluorescence. Following incubation, cells were washed with cold 1x PBS, fixed for 10 min in 2% paraformaldehyde and their green MFI recorded by flow cytometry. ANA-1 Mφ cells were used as a positive control.

## May-Gruenwald-Giemsa staining

Cells were washed, cytospun at 500 g for 8 min onto glass slides and air-dried. The slides were stained with May-Gruenwald stain (Merck, diluted 1:2 at pH 6.8) for 5 min, washed two times with warm $H_2O$ by immersion, followed by staining with Giemsa stain (Merck, diluted 1:10) for 20 min, then washed briefly with warm $H_2O$. Subsequently, slides were mounted, and images acquired using a x100 objective with immersion oil.

## Flow cytometry and fluorescence-activated cell sorting

Cell surface antigens were detected after incubation with FcR blocking reagent (mouse or human, both Miltenyi Biotec) in staining buffer (1% FBS, 0.04% sodium azide in 1× PBS) using fluorescence- or biotin-conjugated antibodies against mouse and human antigens as listed in Supplementary Data 6. Biotinylated antibody-labeled samples were secondarily stained with Streptavidin-Allophycocyanin (Miltenyi Bio-tec). Cell-bound fluorescence signals on $2 \times 10^4$ cells per staining were acquired on a FACSCalibur flow cytometer (BD Biosciences), except for Supplementary Fig. 2d, Supplementary. Fig. 10c on a FACSCanto

(BD) and for Figs. 4f, 8, Supplementary Figs. 6c, d, 9c, 10b and 11b on a Guava easyCyte cytometer. Cell debris and dead cells were omitted from analysis based on scatter signals and gating on propidium iodide-negative cells. Data was analyzed with FlowJo V10.2 software (FlowJo, LLC) and gates were set according to isotype control antibody-stained samples. Correction for background staining or autofluorescence was performed by subtracting isotype control percentage or MFI values from specifically stained sample values. FACS by an S3e cell sorter (Bio-Rad) was used to isolate GFP- or DsRed-positive cells, or to separate ADR-exposed LC based on CD115 expression after labeling with mouse CD115-biotin (Miltenyi Biotec) and CD19-AlexaFluor647 (Biolegend) antibodies, followed by Streptavidin-PE.

DLBCL cell lines were labeled with the CellTrace™ Far Red Cell Proliferation Kit (Thermo Fisher Scientific) one day after treatment with ADR. Five days following treatment (or seven days for RC-K8), ADR treated specimens were sorted for cell proliferation marker retaining CellTrace^high (and for CellTrace^low as control), and untreated controls for CellTrace^low fractions.

See Supplementary Figs. 5b and 12 for gating strategies.

## T-cell killing assays
Spleens from C57BL/6 N mice, twice (7 days apart) intravenously inoculated with strain-matched lymphoma cells, were lysed two weeks after initial inoculation using the gentleMACS system (Miltenyi Biotec). After red blood cell lysis, T-cells were isolated by MACS using mouse Pan T cell isolation kit II and further separated by mouse CD4$^+$ T Cell Isolation Kit or mouse CD8a$^+$ T Cell Isolation Kit (all Miltenyi Biotec). These T-cells were co-cultured in triplicates with GFP-tagged mouse LC at an effector:target cell ratio of 5:1 to 8:1, in the presence or absence of additional antibodies (InVivoMAb anti-mouse PD-L1 [B7-H1] at 1 µg ml$^{-1}$; InVivoMAb anti-mouse MHC Class II [I-A/I-E] at 5 µg ml$^{-1}$; both from BioXCell). After 72 h, the number of viable, GFP$^+$ lymphoma cells was cytometrically determined (see 'Analysis of cell viability' section). The co-cultures were also stained with Ghost Dye™ Violet 450 (Tonbo biosciences) and a panel of T-cell directed antibodies (CD3e, CD4, CD8a, CD69, CD134, see Supplementary Data 6) and fixed before acquisition on a Guava easyCyte 12HT cytometer. For some experiments, primed T-cells were short-term expanded via Dynabeads™ Mouse T-Activator CD3/CD28 for T-Cell Expansion and Activation (Gibco) and 30 U ml$^{-1}$ of rm IL-2 (Sigma) in T-cell medium (as in[96]). In these cases, co-cultures with lymphoma cells were analyzed for viability after 48 h, and equally expanded, naïve T-cells were used as controls.

Lymphomas transduced with 4-OHT-inducible PU.1-ER$^{T2}$ or C/EBPβ-ER$^{T2}$ were first exposed to ADR or left untreated. After 3 days they were further treated with 4-OHT or solvent (EtOH) for 4 days. 7 days after start of ADR treatment they were co-cultured with T-cells in the maintained presence of 4-OHT or solvent.

## Capillary-based immunoassay
Cells were harvested and lysed to extract nuclear, cytoplasmic, or whole-cell protein using the Nuclear Extract Kit (Active Motif) according to the manufacturer's instructions. Protein concentrations were determined using the Pierce™ BCA Protein Assay Kit (Thermo Fisher Scientific) and Pierce™ BSA Standard (Thermo Fisher Scientific). Protein separation and antigen detection were performed using an automated capillary protein electrophoresis system (Simple Western Wes, ProteinSimple) with the 12-230 kDa Wes Separation Module (8 ×25 capillary cartridges) according to supplier's recommendations. 4 µg or 1 µg of lysates were loaded for detection of antigens of interest or loading controls, respectively. Primary antibodies for detection are listed in Supplementary Data 6. Default settings were used for separation, except for the primary antibody incubation time, which was 120 min for PU.1, C/EBPβ, Pax5, M-CSFR and p-c-Jun antibodies. Signals were quantified as areas

under the curve using the 'Compass for SW' software (ProteinSimple) after appropriate adjustment of baseline points. The Compass software automatically computed molecular weights of called peaks and visualized signals.

## Immunoblot
Lysates prepared as described above were separated by SDS-PAGE, transferred onto a PVDF membrane and probed with specific antibodies as listed in Supplementary Data 6. Bands were detected using Immobilon Western HRP (MerckMillipore) on a ChemiDoc MP Imaging System (Bio-Rad) as previously described[54].

## Gene expression analysis by quantitative reverse-transcription PCR
RNA was extracted, cDNA synthesized, and quantitative reverse transcription PCR (qRT-PCR) was performed as previously described[25,54]. Mouse and human transcripts were probed using commercially available Taqman assays (listed in Supplementary Data 6) and Biozym Probe qPCR Kit (Biozym) with the exception of *Cebpb* for which custom-made primers (*Cebpb*-fwd 5′-GGTTTCGGGACTTGATGCA-3′, *Cebpb*-rev 5′-CAACAACCCCGCAGGAAC-3′[97]; *Gapdh*-fwd 5′-CTCCCACT CTTCCACCTTCG-3′, *Gapdh*-rev 5′-GCCTCTCTTGCTCAGTGTCC-3′[98]) and Biozym Blue S'Green qPCR Kit (Biozym) were used. *Gapdh* for mouse and *RPL19* for human samples served as housekeeping genes to determine ΔCt values and the $2^{\Delta\Delta Ct}$ method was used to calculate expression levels relative to respective reference samples. See Source Data for individual qRT-PCR experiment fold-change values that were used to heatmap-visualize data.

## RNA-seq and differential gene expression analysis
shRNA transduced, TIS and non-TIS LC were purified from dead cells by Dead cell removal Kit (Miltenyi Biotec) before RNA extraction with RNeasy Plus Mini Kit (Qiagen). RNA sequencing (RNA-seq) was performed by the DKFZ High Throughput Sequencing Facility (Heidelberg, Germany) on a HiSeq 4000 system (Illumina) with single-read 50 bp specification. Raw reads were trimmed using trimgalore (0.6.4_dev) and quality check of raw and trimmed reads was performed with FastQC (v0.11.9) and default parameters. Trimmed reads were quantified using Salmon (1.3.0). Library type was automatically inferred, mappings were validated, and the bias correction options seqBias and gcBias were used. The index was built from Mus_musculus.GRCm38.cdna.all.fa.gz from Ensembl using a kmer-size of 21. Quantifications were read into R using the tximeta (1.6.3) package and summarized to gene level. Differential expression (DE) analysis was carried out by DESeq2 (1.28.1), excluding all genes with no counts after size factor estimation. P value adjustment was performed by the Benjamini-Hochberg method[99]. Transcripts were considered differentially regulated between shRNA conditions (n = 4 for *Suv39h1*$^{+/+}$ or n = 2 for *Suv39h1*$^-$ genotypes) if |log$_2$ fold-change| ≥ 0.58 and adjusted P value ≤ 0.10. Volcano plots were generated via R scripts provided by the Galaxy platform[100]. DE analysis results can be visualized at https://vcg. github.io/upset/ by providing the following URL as input JSON file via 'Load Data': https://raw.githubusercontent.com/quickaccessfor/ projectrnaseqanalysis/main/rnaseq_shTF_description_fc0.58.json.

RNA-seq of 18 DLBCL cell lines that were separated into ADR-treated CellTrace^high and UT CellTrace^low cell fractions (cf. Supplementary Data 4) was performed as above but with paired-end 100 bp specification. The generated data was processed as described above, except Salmon (v1.5.2) was used, no bias correction was undertaken, and pre-built version of the hg19 index produced with salmon was used, applying a selective alignment method (kmer-size of 31). Subsequently, summarized gene-level estimates were imported using tximport R package (v1.24.0). DE analysis was performed as described above.

## Single-cell RNA-sequencing and analysis

scRNA-seq data was obtained on a 10x Genomics Chromium platform via Illumina NextSeq550 paired-end 75 bp sequencing from *Suv39h1⁻;bcl2;Suv39h1*-ER^T2 LC 5-day exposed to ADR ± 4-OHT[16]. Raw reads for each sample were processed using the 'count' command of Cell Ranger software v7.10. Reads were aligned to the mouse mm39 (GRCm39) genome. The generated report was used to assess sample quality and reads were confidently mapped. 3879 non-TIS and 1395 TIS cells were retained meeting the qualification of more than $10^3$ expressed genes and less than 5% mitochondrial RNA transcripts for subsequent analysis using Seurat (4.3.0) package[101]. Genes expressed in fewer than four cells were removed. The merged dataset was normalized using 'LogNormalize'. The top $2 \times 10^3$ most variable genes were calculated using 'FindVariableFeatures', and principal component analysis was performed on them using 'RunPCA'. Subsequently, the first 10 PCA dimensions were used for t-distributed stochastic neighbor embedding (tSNE) analysis using 'RunTSNE'. Gene set enrichment for each cell was computed by AUCell (1.20.1)[102]. For color-coded tSNE projection, the AUC scores were first $Z$-scored, winsorized and then scaled to a range of 0 to 1. Expression of B-cell transcripts was projected as the average count of the 5 B-cell genes *Ikzf1, Tcf3, Ighm, Ptprc* and *Blnk*.

The raw data for the scRNA-seq dataset of mouse embryonic fibroblasts (MEF) PRJCA002591[103] was obtained from China National Center for Bioinformation (CNCB, https://ngdc.cncb.ac.cn/gsa/browse/CRA002582). Raw reads for each sample were processed using the --soloType argument of STAR software v2.7.10b. Reads were aligned to the mouse reference GRCm39 genome. The generated report was used to assess sample quality and reads were confidently mapped. 171 cells were retained meeting the qualification of more than 1000 expressed genes and less than 5% mitochondrial RNA transcripts for subsequent analysis using Seurat (4.3.0) package. Genes expressed in fewer than four cells were removed. The dataset was normalized using 'SCTransform'. The top 5000 highly variable genes were selected to obtain the first 10 dimensions from principal component analysis, subsequently used for tSNE analysis. Gene set enrichment for each cell was computed by AUCell (1.20.1).

## ATAC-sequencing and analysis

DNA from nuclei of two independent *Suv39h1⁻;bcl2;Suv39h1*-ER^T2 LC 5-day exposed to ADR ± 4-OHT, or left untreated, was extracted, libraries prepared and sequenced as previously described[25].

Paired-end reads with quality score higher than 30 were selected and adapters were trimmed using the Fastq-Mcf software[104]. Sample quality was inspected with FastQC[105]. Reads were aligned to the mm10 version of the *Mus musculus* genome using the bowtie2 software[106], PCR duplicates were eliminated with PicardTools and reads mapping to ENCODE exclusion list and mitochondrial genome were removed. Accessibility peaks were identified with MACS2[107] and reproducible peaks across replicates were selected using the Irreproducibility Discovery Rate (IDR) tool[108].

TF chromatin binding was inferred using the HINT tool[109] and the closest genes to AP-1, CEBPβ and PU.1 were identified with the R package ChIPseeker[110]. For each TF, a hypergeometric test was performed to compute the significance of the overlap between their identified targets and seven gene list signatures, namely: downregulated genes under JUN inhibition[25]; differentially expressed genes under C/EBPβ overexpression or inhibition[49]; genes activated after PU.1 transfection[50]; genes downregulated after PU.1 inhibition (present paper); and genes upregulated in either Mφ, DC or neutrophils relative to B cells[56].

## Microarray data processing and analysis

To compare Eμ-*myc* lymphomas to normal murine C57BL/6 matched GCB, CEL-Files were downloaded from NCBI GEO, entry GSE134751 (sequential numbers GSM3966883 - GSM3966892; GSM3966923 - GSM3966932) and from GSE94733 (sequential numbers GSM2481692 - GSM2481703), both acquired on the Affymetrix Mouse Gene 1.0 ST Array platform. Raw data CEL files from the latter two and the GSE31312 data set (International DLBCL Rituximab-CHOP Consortium Program Study) were preprocessed and normalized using the Bioconductor R oligo package with the function 'rma' as previously described[39]. The 19 probes that had yielded 'NA' values for all samples in GSE31312 were removed from further processing and data analysis, and the remaining 'NA' values were substituted with '0', affecting only 7 data points in subsequent downstream analyses.

## Gene set enrichment analysis

Gene set enrichment analysis (GSEA) was performed with GSEA Desktop v4.0 software (Broad Institute, Inc.) as described[48], using 1000 permutations and "gene set" as permutation type, except for the Duke DLBCL data set[68], where "phenotype" permutation was selected. For the RNA-seq data sets GSE102639, GSE165532, GSE141454 and the Eμ-*myc*;*bcl2*;sh_TF data set (GSE224948), transcripts were excluded from GSEA for which less than half of the samples had reads. Signal2Noise ranking metric was used when three or more samples, and ratio of classes when less than three samples per group were available. Custom signatures, as listed in Supplementary Data 7, and selected MSigDB signatures were probed along with hallmark gene sets (h.all.v7.1.symbols.gmt). Specifically, differential gene sets between murine (microarray data) and human hematopoietic lineages (RNA-seq data) were retrieved from https://www.haemosphere.org[56], or from respective publications. Enrichments were considered significant if |normalized enrichment score| (NES) $\geq 1.3$, nominal $P < 0.05$ and false discovery rate (FDR) $Q < 0.05$ ("gene set" permutation) or $Q \leq 0.25$ ("phenotype" permutation).

## Data set retrieval

The microarray-based lymphoma senescence data set from control;*bcl2* and Suv39h1⁻;*bcl2* LC was obtained from GSE134753[39]; gene expression data of Eμ-*myc* lymphomas propagated and subjected to CTX therapy in vivo in a clinical-trial like fashion was obtained from GSE134751[39]; analogous data from LN preparations of a subset of the same, clinical course annotated Eμ-*myc* lymphomas, purified by CD19 microbeads, (mouse; Miltenyi Biotec) was generated by RNA-seq; RNA-seq data of lymphomas formed from stably MyD88-L265P or empty vector transduced Eμ-*myc* transgenic hematopoietic stem cells was obtained from GSE141454[6] (sequential accession identifiers GSM4203328 – GSM4203334, GSM4203343 - GSM4203349). Further senescence studies GEP were retrieved, including data from RNA-seq of ex vivo, daunorubicin treated primary patient AML cells undergoing TIS (current study); from GSE102639 (Aurora kinase-inhibitor treated A549 lung cancer cells; samples GSM2742113 - GSM2742120), GSE133683 (in vivo doxorubicin treated, primary MMTV-Wnt1 mammary tumors; samples GSM3914423 - GSM3914434), GSE46801 (Braf ^V600E infected, primary human melanocytes; samples GSM1138530 - GSM1138538), GSE242433 (*Pten* knockout in murine prostate epithelium in vivo; samples GSM7763489 - GSM7763498; raw reads were normalized using DESeq2) and GSE165532 (Hras^G12V-transduced or retrovirus-infected Wi38 HDF; samples GSM5035467 - GSM5035478). The Duke DLBCL cohort RNA-seq data set[68] was retrieved from the European Genome-Phenome Archive (https://ega-archive.org) via the identifier EGAD00001003600 and transcripts quantified using salmon and imported using tximeta and tximport packages. Copy number alteration data for the Duke cohort was used as provided[68]. The mutation call data for the Duke DLBCL cohort was retrieved from the re-analysis by the GAMBL consortium[111]. The series matrix file of the Lenz et al. DLBCL data set GSE10846[72] was retrieved as provided by NCBI GEO. The GSE2350 data set was used to compare normal and malignant human B-cells (CD19⁺ DLBCL samples with sequential identifier numbers GSM44246 - GSM44252 and normal GCB centroblasts GSM44143 - GSM44147, GSM65674 - GSM65677, GSM65680).

## Annotation analysis of differentially expressed genes

Induced network module analysis[47] was performed on the CPDB website (http://cpdb.molgen.mpg.de/), using gene-regulatory interactions and allowing for intermediate nodes. TIS-specific transcripts in the Eμ-myc;bcl2 model retrieved as previously described[25,39] were supplied (Supplementary Data 1).

Pathway overrepresentation analysis was conducted using default parameters with the 'Data Analysis' option of the innateDB platform (https://innatedb.com)[78].

## TF footprinting analysis

Differentially expressed genes between senescence-capability categories of DLBCL cell lines were read into the Bioconductor R DoRothEA package[67] and differential TF activities (dTF) as normalized enrichment scores (NES) were computed using dorothea hs_regulons of A, B and C confidence and the DESeq2 Wald statistic as quantitative input. To estimate the differential shifts in TF activity caused by ADR treatment, ddTF scores were computed by the formula ddTF = dTF[category1$_{ADR}$ vs. category2$_{ADR}$] – dTF[category1$_{UT}$ vs. category2$_{UT}$]. Volcano plots were generated via the 'volcano_nice' function.

## Immune deconvolution

Immunedeconv R package (v 2.0.4) was used for unified access to computational methods for estimating immune cell fractions from bulk RNA sequencing data (xCell, quanTIseq and CIBERSORT). Estimated contributions of myeloid cell types were aggregated to give total contribution values of myeloid cells ("myeloid score") to apparent sample composition as summarized in Supplementary Data 2.

## Lymphoma ecotype and LymphGen classification

Recovery of Bcs, TMEcs and LE for Duke DLBCL and Eμ-myc clinical-trial like data sets was computed using R implementation of the EcoTyper software as described in EcoTyper GitHub repository (https://github.com/digitalcytometry/ecotyper)[76]. LymphGen classification of the Schmitz et al., DLBCL cohort[75] was retrieved from the respective publication[73].

## Survival analysis and gene signature score analysis

For Kaplan-Meier survival analyses based on the average expression of gene signatures, the data sets GSE134751, GSE31312, Duke DLBCL, and GSE10846 were Z-score scaled per row (probe/transcript) and data points winsorized to a |Z|-value of 2.5. Kaplan-Meier plots were then generated with the Bioconductor R SigCheck package using the function 'sigCheck', with patients stratified as above and below the median of gene signature scores. Gene signature scores were calculated per sample as the arithmetic mean of transcript Z-scores within the gene signature. P values were calculated by the log-rank test.

## Statistical evaluation and graphical display

Unless stated otherwise, data are presented as arithmetic means ± standard error of the mean (SEM) and statistical analyses were based on paired or unpaired two-sided t-tests with $P < 0.05$ considered statistically significant. No correction for multiple hypothesis testing was undertaken, except in analyses of high-throughput data sets as indicated in figure legends. No statistical method was used to predetermine sample size. The investigators were not blinded to allocation during experiments and outcome assessment. One ex vivo T$_{UT}$ T-cell isolate (relates to Fig. 8b, c, e–h) was excluded from being evaluated on TIS lymphoma cells, since it contained a small remaining lymphoma cell fraction that quickly overgrew the cell cycle-arrested senescent cultures. Statistical analysis was performed using GraphPad v9 or R package stats. Linear regression of CD115-separated protein or gene expression data, as displayed in Fig. 2b, c, was performed using a fixed-effects model with the 'lm' function from the R stats package. The analysis was based on log$_2$-transformed fold-change values to ensure a normal distribution of residuals. In the model, 'lymphoma' and either 'protein' (for Fig. 2b) or 'gene' (for Fig. 2c) were used as covariates, with an interaction term between 'protein' and CD115 status included in the model for protein data (Fig. 2b; formula = lm(log$_2$fold-change ~ 0 + lymphoma + cd115sort * protein)). For gene expression data (Fig. 2c), a separate model for each transcript class and excluding the Csf1r gene was used (formula = lm(log$_2$fold-change ~ 0 + lymphoma + cd115sort + gene)). Plots were generated using GraphPad or R packages ggplot2 and ggpubr. Box plots consist of the median as center line, 25% and 75% percentiles as box limits and whiskers extending to largest and smallest values within the 1.5× interquartile range of the box limits. Heatmaps were generated by the Heatmapper online tool (http://heatmapper.ca) or the R pheatmap package. The image of mouse in Fig. 8a is adapted from Servier Medical Art (https://smart.servier.com/smart_image/mouse-3/) licensed under a Creative Commons Attribution 3.0 Unported License. Inkscape software v1.2 was used to format graphs.

## Reporting summary

Further information on research design is available in the Nature Portfolio Reporting Summary linked to this article.

## Data availability

RNA-seq data generated in this study have been deposited at the Gene Expression Omnibus (GEO) repository of the National Center for Biotechnology Information (NCBI) under primary accession numbers GSE224948 (RNA-seq of Eμ-myc;bcl2;sh_TF lymphoma cells – Fig. 3d, Fig. 7a, Supplementary Fig. 4), GSE224867 (RNA-seq of DLBCL cell lines – Fig. 4d, Supplementary. Figure 5), GSE232445 (scRNA-seq data of Suv39h1-inducible lymphoma cells – Fig. 1d), GSE289631 (RNA-seq of CD19$^+$ selected, outcome annotated Eμ-myc lymphomas – Fig. 5c, d) and GSE289427 (RNA-seq of ex vivo daunorubicin treated AML blasts – Supplementary Fig. 1c). ATAC-seq data of Suv39h1-inducible lymphoma cells (Supplementary Fig. 4g) are hosted on the Sequence Read Archive (BioProject no. PRJNA1201841). Referenced data sets are available under following accession codes: GSE134753 – Eμ-myc;bcl2 - TIS model. GSE134751 – Eμ-myc - clinical trial like study. GSE141454 – Eμ-myc;-MyD88-L265P GEP. GSE94733 – normal mouse GCB cells. GSE2350 – includes CD19$^+$ selected DLBCL and normal GCB. GSE31312 – GEP of DLBCL patients from The International DLBCL Rituximab-CHOP Consortium Program. EGAD00001003600 – Duke DLBCL cohort GEP. GSE10846 – GEP of DLBCL patients from the Lenz/Staudt cohort. dbGAP phs001444.v2.p1 – Schmitz/Staudt DLBCL GEP. GSE102639 – Aurora kinase-inhibitor treated A549 lung cancer cells. GSE133683 – includes in vivo doxorubicin treated, primary MMTV-Wnt1 mammary tumors with wt p53. GSE46801 – Braf$^{V600E}$ infected, primary human melanocytes. GSE165532 – Hras$^{G12V}$-OIS and VIS of Wi38 HDF. GSE242433 – Pten knockout in murine prostate epithelium. CRA002582 – scRNA-seq data of 175 MEF cells at different stages of a senescence trajectory. GSE61608 – GEP of B-cells activated by B-cell receptor stimulation or lipopolysaccharide. GSE35998 – GEP of B-cells activated by lipopolysaccharide and CD40 ligand. GSE16874 – GEP of B-cells activated by lipopolysaccharide and IL-4. The data generated in this study and displayed in charts and graphs are available in the Source Data file or in Supplementary Data 1-5 whenever source data were not already included in the display item. Source data are provided with this paper.

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

## Acknowledgements

We thank Achim Leutz for materials and members of the collaborating labs for discussions and editorial advice, and Andreas Sommer (Vienna BioCenter Core Facilities) for running scRNA-seq. This work was supported by grants to C.A.S. from the Deutsche Krebshilfe (No. 7011377629), the Deutsche Forschungsgemeinschaft DFG (GO 2688/1-1 | SCHM 1633/11-1, SCHM 1633/9-2), the European Union ERA PerMed program (within the 'HiRisk-HiGain' consortium) via the German Federal Ministry of Health (BMG), and the Förderverein Hämatologie und internistische Onkologie (Tyle Private Foundation, Linz, Austria), to S.L. and C.A.S. from the Johannes Kepler University Linz, Austria (SeedPlusMed grant 'SeeDoEnable'), and to B.C., S.L., U.L., J.W. and C.A.S. by the German Federal Ministry of Education and Reseach (BMBF) e:Med program project SeneSys (No. 031L0189A). This interdisciplinary work was further made possible by the Berlin School of Integrative Oncology (BSIO) graduate program funded within the German Excellence Initiative (with D.B. and X.C. as members of this program), and the German Cancer Consortium (GCC).

## Author contributions

D.B. performed flow cytometry, and cytochemical assays, and with help from A.L., and K.P. in vitro mouse lymphoma work, molecular genetics and cell biological experiments. D.B. and G.K. conducted analyses with human DLBCL cell lines and acquired RNA-seq data. P.R.-P., Z.S., D.B., T.P.H., and N.T. processed raw data and carried out bioinformatic analyses. A.L., J.K., X.C., A. Bittner and B.B. carried out mouse experiments. E.S. and Y.Y. conducted OIS experiments. A. Bhattacharya, M.S. and D.N.Y.F. conceptualized and obtained RNA-seq data of primary tumor material. M.M. and D.N.Y.F. acquired and Z.S. analyzed scRNA-seq data, and J.A.N.L.F.F. and O.B. analyzed ATAC-seq data. D.B. analyzed data and prepared figures. M.M., D.B., M.R., J.R.D., A. Bhattacharya, D.N.Y.F., X.C., J.K., J.D., B.C., S.L., U.L., J.W. and C. Scheidereit contributed to study design and data interpretation. All authors discussed the results and contributed to the final manuscript. C.A. Schmitt designed the project, supervised the data analysis and wrote the manuscript.

## Funding

## Competing interests

The authors declare no competing interests.

## Additional information

[1]Charité - Universitätsmedizin, corporate member of Freie Universität Berlin, Humboldt-Universität zu Berlin, and Berlin Institute of Health, Department of Hematology, Oncology and Tumor Immunology, and Molekulares Krebsforschungszentrum – MKFZ, Campus Virchow Klinikum, Berlin, Germany. [2]Max Delbrück Center for Molecular Medicine in the Helmholtz Association, Berlin, Germany. [3]IMRB, Mondor Institute for Biomedical Research, INSERM U955 – Université Paris Est Créteil, UPEC, Faculté de Médecine de Créteil, Créteil, France. [4]Knowledge Management in Bioinformatics, Institute for Computer Science, Humboldt-Universität zu Berlin, Berlin, Germany. [5]Johannes Kepler University, Medical Faculty, Linz, Austria. [6]Department of Oncology, Hematology

and Bone Marrow Transplantation with Section of Pneumology, University Medical Center Hamburg-Eppendorf, Hamburg, Germany. [7]Research Institute of Molecular Pathology (IMP), Vienna BioCenter (VBC), Vienna, Austria. [8]Experimental Pharmacology & Oncology Berlin-Buch GmbH, Berlin, Germany. [9]Core Unit Bioinformatics – CUBI, Berlin Institute of Health, Berlin, Germany. [10]Charité - Universitätsmedizin, corporate member of Freie Universität Berlin, Humboldt-Universität zu Berlin, and Berlin Institute of Health, Department of Pediatric Oncology and Hematology, Charité-Universitätsmedizin Berlin, Berlin, Germany. [11]Charité - Universitätsmedizin, corporate member of Freie Universität Berlin, Humboldt-Universität zu Berlin, and Berlin Institute of Health, Experimental and Clinical Research Center (ECRC) of the Max Delbrück Center for Molecular Medicine and Charité-Universitätsmedizin Berlin, Berlin, Germany. [12]Charité - Universitätsmedizin, corporate member of Freie Universität Berlin, Humboldt-Universität zu Berlin, and Berlin Institute of Health, Department of Hematology, Oncology, and Cancer Immunology, Campus Benjamin Franklin, Berlin, Germany. [13]Deutsches Konsortium für Translationale Krebsforschung (German Cancer Consortium), partner site Berlin, Berlin, Germany. [14]Medical Research Center and Department of Oncology Binzhou Medical University Hospital, 256600 Binzhou, P.R. China. [15]Department of Mathematics and Computer Science, Free University Berlin, Berlin, Germany. [16]Kepler University Hospital, Department of Hematology and Oncology, Krankenhausstraße 9, 4020 Linz, Austria. ✉e-mail: clemens.schmitt@charite.de; clemens.schmitt@kepleruniklinikum.at

