## [Transparent Peer Review file · Nature Communications]

Senescence-associated lineage-aberrant plasticity evokes T-cell-mediated tumor control

Corresponding Author: Professor Clemens Schmitt

Version 0:

Reviewer comments:

Reviewer #1

(Remarks to the Author)

The revised version of the manuscript addresses most reviewers' comments and is now much improved. A change in terminology addressed the main concern regarding the lack of evidence of transdifferentiation of senescent B cells into myeloid-like cells—the new term “cross-differentiation” better reflects the findings. The manuscript was also strengthened by additional data showing a similar pattern lineage lineage-induced plasticity occurring beyond B cell lymphomas.

Reviewer #2

(Remarks to the Author)

Although the authors did not provide experimental data or additional analysis for every point raised, they made a thorough effort to address all the concerns. Their responses were thoughtful and comprehensive, demonstrating a commitment to improving the manuscript based on the feedback. The revisions have significantly strengthened the paper, and it is now suitable for publication in its current form.

Reviewer #3

(Remarks to the Author)

The authors have addressed my questions adequately. I have no further comments.
